# Counterfactual Contrastive Learning with Normalizing Flows for Robust Treatment Effect Estimation

Jiaxuan Zhang [1]   Emadeldeen Eldele [2]   Fuyuan Cao [1]   Yang Wang [1]   Xiaoli Li [2 3]   Jiye Liang [1]

## Abstract

Estimating Individual Treatment Effects (ITE) from observational data is challenging due to covariate shift and counterfactual absence. While existing methods attempt to balance distributions globally, they often lack fine-grained sample-level alignment, especially in scenarios with significant individual heterogeneity. To address these issues, we reconsider counterfactual as a proxy to emulate balanced randomization. Furthermore, we derive a theoretical bound that links the expected ITE estimation error to both factual prediction errors and representation distances between factuals and counterfactuals. Building on this theoretical foundation, we propose FCCL, a novel method designed to effectively capture the nuances of potential outcomes under different treatments by (i) generating diffeomorphic counterfactuals that adhere to the data manifold while maintaining high semantic similarity to their factual counterparts, and (ii) mitigating distribution shift via sample-level alignment grounded in our derived generalization-error bound, which considers factual-counterfactual similarity and category consistency. Extensive evaluations on benchmark datasets demonstrate that FCCL outperforms 13 state-of-the-art methods, particularly in capturing individual-level heterogeneity and handling sparse boundary samples.

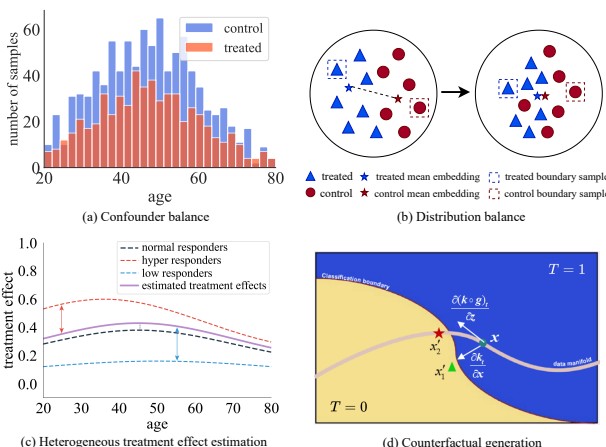

*Figure 1.* (a) Existing balance methods align the overall distributional statistics (e.g., mean) of confounders (e.g., age) between treated and control groups. (b) Maximum Mean Discrepancy (MMD) minimizes group-level discrepancies by aligning mean representations but fails to capture individual-level heterogeneity, particularly for boundary samples. (c) Current methods effectively estimate ITE for normal responders but result in higher errors for hyper and low responders, highlighting the need for finer-grained alignment. (d) $x_1'$ is generated via gradient ascent until the treatment label flips from $T = 1$ to $T = 0$, while $x_2'$ represents the diffeomorphic counterfactual generated in the latent space of the normalizing flow, ensuring adherence to the data manifold.

## 1. Introduction

Estimating the Individual Treatment Effects (ITE) from observational data is critical for personalized decision-making in diverse domains such as digital marketing (Li et al., 2024;

[1]School of Computer and Information Technology, Shanxi University, Taiyuan, China [2]Institute for InfoComm Research, Agency for Science, Technology and Research, Singapore [3]Centre for Frontier AI Research, Agency for Science, Technology and Research, Singapore. Correspondence to: Fuyuan Cao <cfy@sxu.edu.cn>.

*Proceedings of the 42$^{nd}$ International Conference on Machine Learning*, Vancouver, Canada. PMLR 267, 2025. Copyright 2025 by the author(s).

Chu et al., 2022), social sciences (Martínez-Sánchez et al., 2024), and healthcare (Liu et al., 2021; Bica & Van der Schaar, 2022). Accurate ITE estimation guides crucial decisions by quantifying the causal impact of interventions. Unlike randomized controlled trials (RCTs), observational studies often suffer from a non-random treatment assignment mechanism, leading to covariate shift (Cheng et al., 2024; Yao et al., 2021; Kong et al., 2023). Accurate estimation of treatment effects requires mitigating this *covariate shift* and predicting *counterfactual outcomes*—estimations of what would have happened to an individual under a different treatment. Nevertheless, generating realistic counterfactuals and achieving a balanced representation remain significant challenges.

Covariate balance methods aim to mitigate distribution shift by aligning the distributions of the treated and control groups (Figure 1a). This helps balance out the influence of confounders (factors that can affect both the treatment and the outcome). Methods like Maximum Mean Discrepancy (MMD) (Jiang & Sun, 2022) minimize the distribution discrepancies by aligning the mean embeddings of the treated and control groups. While generally effective, these methods neglect individual-level heterogeneity (Wu et al., 2023), particularly for boundary samples (Figure 1b). This oversight undermines ITE estimation accuracy and confidence, especially in scenarios with sparse or extreme samples that deviate significantly from population averages.

For example, in precision medicine, representation learning methods may balance confounder distributions like age (which influences both medication allocation and health outcomes), but fail to capture critical variations in individual drug sensitivity. Patients typically fall into three response categories: normal responders, hyper-responders, and low responders (Figure 1c). While current methods may perform adequately for the majority (normal responders), they frequently produce significant errors for hyper-responders, who risk adverse effects, and low responders, who may require an increased interventions dosage (Feuerriegel et al., 2024). This lack of fine-grained alignment can compromise treatment decisions for these vulnerable groups.

To address sample-level heterogeneity, recent approaches emphasize aligning pairs of samples, particularly those in intermediate or boundary regions identified through propensity scores (Yao et al., 2018; Li & Yao, 2022; Zhao et al., 2024). While promising, these methods achieve partial alignment and still face challenges in effectively mitigating distribution shifts. Generating instance-specific counterfactuals offers a potential solution, as these proxies emulate randomized treatment assignment by constructing personalized comparisons. However, this strategy is highly sensitive to noise (Cai et al., 2024), often producing adversarial examples rather than meaningful counterfactuals (Figure 1d).

To overcome these limitations, we propose FCCL (**F**low-based **C**ounterfactual **C**ontrastive **L**earning), a novel framework that integrates two key techniques: (1) Flow-based Counterfactual Generation, which utilizes normalizing flows to generate realistic counterfactuals that adhere to the data manifold while minimally altering sample semantics. This ensures that counterfactuals maintain fidelity to real-world plausibility while reversing treatment classification. (2) Sample-level Alignment for Balancing Distribution, which employs contrastive loss to optimize the representation distances between factual and counterfactual samples based on category consistency and similarity.

This dual strategy in FCCL enables robust sample-level alignment, effectively capturing the nuances of potential outcomes under different treatments and enhancing ITE estimation, particularly for boundary and heterogeneous samples (see Figure 3). Furthermore, we provide a novel generalization-error bound that links ITE estimation error to factual prediction error and the representation distance between factual and counterfactual samples, validating FCCL's efficacy.

Our main contributions are summarized as follows:

- We introduce a theoretically grounded diffeomorphic counterfactual as a proxy, which ensures counterfactuals are realistic and semantically meaningful by adhering to the data manifold.

- We derive a theoretical ITE generalization-error bound based on factual error and representation distances, and propose FCCL, a novel contrastive learning framework for ITE estimation that aligns individual sample representations across treatments.

- We conduct a comprehensive evaluation of FCCL, demonstrating its superior performance over CFR-based and adversarial training-based methods, with up to 25% and 33% reduction in estimation error on IHDP, achieving more accurate and stable ITE estimation.

## 2. Related Work

Traditional treatment effect estimation methods either rely on propensity score or directly optimize sample weight, including propensity score matching, debiased machine learning, and entropy balancing (Li et al., 2016; Jung et al., 2021; Athey et al., 2018). However, these methods require correct model specification; otherwise, they may lead to unreliable estimates. Recently, numerous representation learning studies focus on learning balanced representations of covariates (Johansson et al., 2022), which can be broadly categorized into covariate balance and adversarial balance approaches.

### 2.1. Covariate Balance

Covariate balance methods aim to learn a representation space $\Phi(\cdot)$ that mitigates treatment selection bias. MMD and the Wasserstein metric were first introduced in TARNet (Johansson et al., 2016) and CFRNet (Shalit et al., 2017) to minimize the distribution differences between treatment and control groups within the representation space, effectively framing causal inference as a transfer learning problem. Notably, CFRNet provides a generalization-error bound, showing that the expected ITE estimation error can be reduced by the difference between the treated and control distributions. These models inspired numerous representation balancing methods, including (Kazemi & Ester, 2024; Zhang et al., 2020). For example, the Perfect Match method (Schwab et al., 2018) extended the TARNet to scenarios with multiple

treatments and addressed varying sample sizes by augmenting samples with propensity-matched nearest neighbors. Huang et al. (2024) derived a new ITE error bound based on $\mathcal{H}$-divergence and proposed DIGNet to mitigate the trade-off problem between the outcome prediction and covariate balance. Besides, DESCAN (Zhong et al., 2022) improved representation learning by jointly learning treatment and response functions over the entire sample space, effectively addressing both selection bias and sample imbalance.

While these methods focus on achieving global distributional balance, they often neglect individual-level information crucial for estimating potential outcomes. SITE (Yao et al., 2018), for instance, attempted to capture sample-level variation by selecting specific sample pairs for alignment. However, these approaches only achieve partial balance and fail to effectively mitigate the distribution shift. This limited alignment prevents robust sample-wise debiasing, which is essential for accurate ITE estimation.

### 2.2. Adversarial Balance

When datasets contain finite samples and complex relationships between covariates, neural network-based feature representations may not be accurate. Adversarial training makes factual and counterfactual distributions indistinguishable, naturally mitigating distribution bias. GANITE (Yoon et al., 2018) utilized adversarial training to prevent the discriminator from distinguishing between true factual outcomes and generated counterfactual outcomes. SCIGAN (Bica et al., 2020) extended GAN architectures with a hierarchical discriminator for factual-counterfactual identification and a multi-task generator, enabling outcome estimation for continuous and multi-level discrete interventions via heterogeneous response curve learning. ABCEI (Du et al., 2021) balanced covariate distributions in the latent space using adversarial training, and addressed information loss with a mutual information estimator. Unlike ABCEI, CBRE (Zhou et al., 2022) introduced an information loop to preserve predictive information that might otherwise be lost during the raw-to-latent space transformation in adversarial training. NCMs (Xia et al., 2022) demonstrated that structural constraints for counterfactual reasoning could be captured and proposed algorithms for joint identification and estimation of counterfactual distributions.

Although adversarial training methods can learn counterfactual distribution, they face significant challenges in generating realistic counterfactuals. Naively adding noise to inputs often produces adversarial examples rather than semantically meaningful counterfactuals. In contrast, our approach generates counterfactuals that preserve semantic integrity, respecting the intrinsic structure of the data manifold. This ensures that the generated counterfactuals are both meaningful and reliable for causal inference tasks.

## 3. Preliminary

### 3.1. Notations and Assumptions

Following the Neyman-Rubin potential outcome framework (Rubin, 2005; Shalit et al., 2017), the covariate space $\mathcal{X}$ is a bounded subset $\mathcal{X} \subset \mathbb{R}^d$, and the potential outcome space is $\mathcal{Y} \subset \mathbb{R}$. The binary treatment indicator $t \in \{0, 1\}$ indicates whether the unit receives treatment $t = 1$ (e.g., medication) or serves as control $t = 0$ (e.g. placebo). We define $y^t$ as the potential outcome (e.g. blood sugar level) for the unit under treatment $t \in \{0, 1\}$, the potential outcomes $y^0, y^1 \in \mathcal{Y}$. Given a unit $x \in \mathcal{X}$ with treatment assignment $t$, we only observe the factual outcome $y^t$, while the counterfactual outcome $y^{1-t}$ remains unobserved. The observed outcome can be expressed as: $y = (1 - t) y^0 + t y^1$. Our goal is to estimate the ITE and evaluate its accuracy using the metric *Precision in the expected Estimation of Heterogeneous Effect* (PEHE).

**Definition 3.1.** The individual treatment effect:
$$\tau(x) := \mathbb{E}\left[y^1 - y^0 | x\right].$$

**Definition 3.2.** Let $f : \mathcal{X} \times \{0, 1\} \to \mathcal{Y}$ by an hypothesis. The estimated individual treatment effect:
$$\hat{\tau}_f(x) = f(x, 1) - f(x, 0).$$

**Definition 3.3.** The expected Precision in Estimation of Heterogeneous Effect loss of $f$:
$$\epsilon_{PEHE}(f) = \int_{\mathcal{X}} (\hat{\tau}_f(x) - \tau(x))^2 p(x) dx. \quad (1)$$

We made the following assumptions to ensure that treatment effects are identifiable:

**Assumption 3.4.** *Consistency, Ignorability, and Overlap.* **Consistency**: For a unit with treatment assignment $t$, the observed outcome equals potential outcome $y^t$. **Ignorability**: The potential outcomes are independent of the treatment conditioning on covariates, such that $(y^1, y^0) \perp\!\!\!\perp t | x$. **Overlap**: For any $x$, the probability of receiving treatment is positive. That is, $0 < P(t = 1|x) < 1$, for $\forall x \in \mathcal{X}$.

### 3.2. Preliminary Propositions

**Assumption 3.5.** *Manifold hypothesis* (Dombrowski et al., 2023). The data is assumed to concentrate within a small region $\delta$ around a submanifold $\mathcal{M}$ of $\mathcal{X}$:
$$\mathcal{S} = \mathcal{M} \times \mathcal{I}_{\delta_1} \times \cdots \times \mathcal{I}_{\delta_{\mathcal{N}_{\mathcal{X}} - \mathcal{N}_{\mathcal{M}}}}, \quad (2)$$

where $\mathcal{I}_{\delta_i} = \left(-\frac{\delta_i}{2}, \frac{\delta_i}{2}\right)$, for $i \in \{1, \ldots, \mathcal{N}_{\mathcal{X}} - \mathcal{N}_{\mathcal{M}}\}$, is an open interval of length $\delta_i$. We assume that $\delta_i$ is small, meaning the data lies approximately on the submanifold $\mathcal{M} \subset \mathcal{X}$, which has a much lower dimensionality $N_{\mathcal{M}}$ than the dimensionality $N_{\mathcal{X}}$ of $\mathcal{X}$.

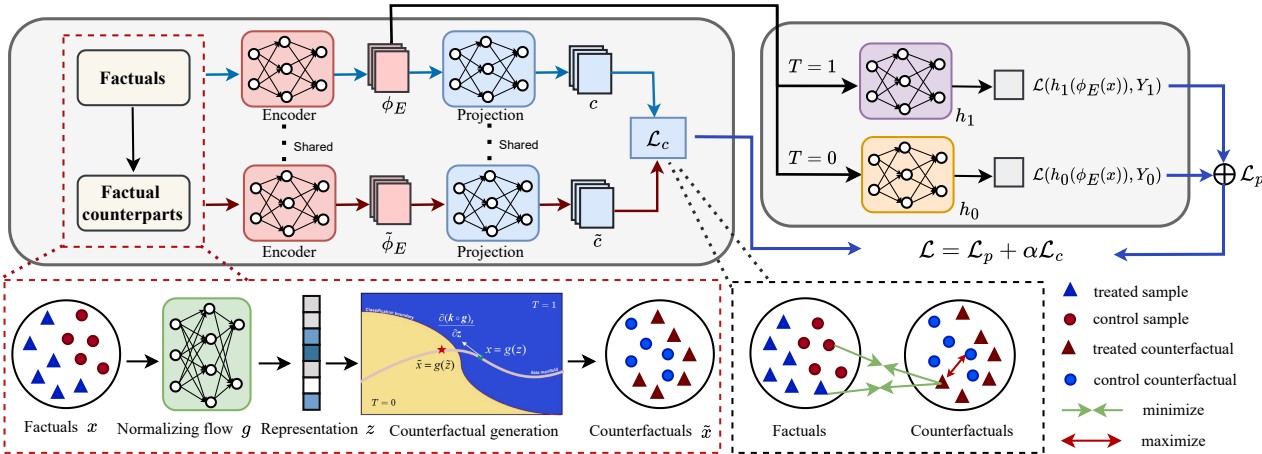

*Figure 2.* The architecture of the proposed FCCL framework. The diffeomorphic counterfactual generation module generates realistic counterfactuals via gradient ascent in the latent space of a normalizing flow, adhering to the data manifold. The contrastive learning module refines sample representations under different treatments, enabling precise instance-level distribution alignment. Finally, two separate neural networks, $h_1(\phi_E(x))$ and $h_0(\phi_E(x))$, are used to estimate potential outcomes under different treatments.

**Definition 3.6. Diffeomorphic counterfactual.** The diffeomorphic counterfactual is defined as minimal modifications on the data manifold of the input reversing the classification decision $k(x)$, represented as:

$$\tilde{x} = \arg\min_{\tilde{x}\in\mathcal{M}} dist(\tilde{x}, x) \ s.t. \ k(\tilde{x}) \neq k(x), \quad (3)$$

where $\mathcal{M}$ is the data manifold, $dist(\tilde{x}, x)$ denotes the distance between $\tilde{x}$ and $x$. The classifier $k : \mathcal{X} \to \{0, 1\}$ assigns an input $x \in \mathcal{X}$ to a class $t \in \{0, 1\}$, and $k(\tilde{x}) \neq k(x)$ indicates a change in the classifier's prediction outcome.

**Definition 3.7.** Let $L : \mathcal{Y} \times \mathcal{Y} \to \mathbb{R}_+$ be the absolute loss or squared loss, $l_{h,\Phi}(x, t)$ denote the expected loss for the unit-treatment pair $(x, t)$, with representation function $\Phi : \mathcal{X} \to \mathcal{R}$ and hypothesis $h : \mathcal{R} \times \{0, 1\} \to \mathcal{Y}$, which can be mathematically represented as: $l_{h,\Phi}(x, t) = \int_{\mathcal{Y}} L(y^t, h(\Phi(x), t))p(y^t|x)dy^t$. The expected factual and counterfactual losses of $h$ and $\Phi$ are:

$$\epsilon_F(h, \Phi) = \int_{\mathcal{X}\times\{0,1\}} l_{h,\Phi}(x, t)p(x, t)dxdt,$$

$$\epsilon_{CF}(h, \Phi) = \int_{\mathcal{X}\times\{0,1\}} l_{h,\Phi}(x, t)p(x, 1-t)dxdt.$$

**Definition 3.8.** The expected factual treated and control losses are:

$$\epsilon_F^{t=1}(h, \Phi) = \int_{\mathcal{X}} l_{h,\Phi}(x, 1)p^{t=1}(x)dx,$$

$$\epsilon_F^{t=0}(h, \Phi) = \int_{\mathcal{X}} l_{h,\Phi}(x, 0)p^{t=0}(x)dx.$$

## 4. Methodology

We propose a novel framework, **FCCL** (**F**low-based **C**ounterfactual **C**ontrastive **L**earning), for individual treatment effect estimation. FCCL is grounded in theoretical insights that justify its key components: (1) flow-based counterfactual generation, (2) contrastive learning for sample-level alignment, and (3) a predictive model for potential outcomes. Figure 2 provides an overview, integrating the theoretical foundation with methodological details of FCCL.

**Motivation.** Randomized controlled trials (RCTs) are recognized as the gold standard for causal inference due to their ability to achieve random treatment assignment independent of covariates (Ma & Zhang, 2023). Our method aims to emulate the balanced randomization inherent in RCTs by employing counterfactual contrastive learning. To the best of our knowledge, *this is the first work to leverage the relationship between factual and counterfactual samples to produce consistent representations that effectively capture the characteristics of potential outcomes under different treatments, thereby addressing distribution shifts between treated and control groups through fine-grained, sample-level alignment.*

### 4.1. Theoretical Foundations

To motivate our methodology, we first establish theoretical bounds on counterfactual loss and ITE estimation error. These bounds guide the design of FCCL and justify its components. Detailed proofs are provided in Appendix A.

**Lemma 4.1.** *Let $\Phi : \mathcal{X} \to \mathcal{R}$ be an invertible representation function with inverse $\Psi$ and $h : \mathcal{R} \times \{0, 1\} \to \mathcal{Y}$ a hypothesis function. Define $u := p(t = 1)$ as the treat-*

*ment proportion. For any treatment indicator $t \in \{0, 1\}$, let $disc(\Phi^t(x), \Phi^t(\tilde{x}))$ represent the distance between counterfactual and factual samples in the representation space. Then, the counterfactual loss $\epsilon_{CF}(h, \Phi)$ is bounded by:*

$$
\epsilon_{CF}(h, \Phi) \leq (1 - u) \cdot \epsilon_F^{t=1}(h, \Phi) + u \cdot \epsilon_F^{t=0}(h, \Phi) + \\
disc(\Phi^t(x), \Phi^t(\tilde{x})) + disc(\Phi^t(\tilde{x}), \Phi^{1-t}(x)),
\tag{4}
$$

*where $\epsilon_{CF}$ and $\epsilon_F$ are defined in Definition 3.7 and Definition 3.8.*

This result highlights the importance of minimizing representation distances between factual and counterfactual samples to reduce counterfactual loss.

**Theorem 4.2.** *Let $\epsilon_{PEHE}(h, \Phi)$ be the estimation error with representation function $\Phi : \mathcal{X} \to \mathcal{R}$ with $\Psi$ being its inverse and hypothesis $h : \mathcal{R} \times \{0, 1\} \to \mathcal{Y}$. For any treatment indicator $t \in \{0, 1\}$, let $disc(\Phi^t(x), \Phi^t(\tilde{x}))$ be the representation distance between counterfactual and factual samples. The error is bounded by:*

$$
\epsilon_{PEHE}(h, \Phi) \leq \\
2(\epsilon_F^{t=1}(h, \Phi) + \epsilon_F^{t=0}(h, \Phi) - 2\sigma_Y^2) + \\
2(disc(\Phi^t(x), \Phi^t(\tilde{x})) + disc(\Phi^t(\tilde{x}), \Phi^{1-t}(x))).
\tag{5}
$$

This bound links ITE estimation error to the generalization error of factual predictions and representation distances, motivating our focus on minimizing these distances.

This theorem proves that the estimation error $\epsilon_{PEHE}(h, \Phi)$ is upper bounded by two terms: the standard generalization error of factual predictions $\epsilon_F$ and the distance constraints in the representation space. These distance constraints represent two types of alignment: (i) between counterfactual samples and their factual counterparts (similarity), and (ii) between counterfactual samples and those from the opposite group of factual samples (category consistency). These alignment errors can be minimized using contrastive loss to achieve sample-wise correspondence in the representation space, improving the accuracy of ITE estimation.

### 4.2. Flow-based Counterfactual Generation

Existing counterfactual generation approaches typically employ gradient ascent optimization in the input space $\mathcal{X}$. Specifically, for a given step size $\eta$ and target class $t$, the iterative gradient update rule is defined as:

$$
x^{(i+1)} = x^{(i)} + \eta \frac{\partial k_t}{\partial x}\left(x^{(i)}\right),
\tag{6}
$$

where the process continues until the classifier $k(x^{(i+1)})_t$ confidence exceeds a threshold $\Lambda$.

However, counterfactuals generated in this manner often result in unstructured perturbations that fail to capture meaningful semantic transformations, frequently deviating from

the underlying data manifold $\mathcal{M}$ that represents the intrinsic structure and variations of the data.

To address this limitation, we generate diffeomorphic counterfactual, ensuring that counterfactual $\tilde{x}$ resides on the same manifold $\mathcal{M}$ as the original instance $x$, i.e., $x, \tilde{x} \in \mathcal{M}$. This is formulated as an optimization problem:

$$
\min D_M(x, \tilde{x}) \quad s.t. \ k(\tilde{x}) \neq k(x),
\tag{7}
$$

where $D_M$ denotes the geodesic distance on the manifold $\mathcal{M}$ and $k(\cdot)$ is a classifier function. By constraining $\tilde{x}$ to remain on $\mathcal{M}$, this formulation avoids unstructured perturbations while flipping the classification outcome.

Normalizing flows provide a principled mechanism to construct such diffeomorphic counterfactuals. In particular, a normalizing flow maps a base distribution $p(z)$ in the latent space $\mathcal{Z}$ to a complex target distribution $p(x)$ via a diffeomorphism $g$: $x = g(z)$. The normalizing flow is trained by minimizing the Kullback-Leibler (KL) divergence between the true data distribution and the model distribution: $\mathcal{L}(\boldsymbol{\theta}) = D_{\mathrm{KL}}\left[p(x) \| p(x; \boldsymbol{\theta})\right]$.

Using the learned flow $g$, counterfactuals are generated by optimizing the composite function $k \circ g : \mathcal{Z} \to \mathcal{Y}$ in the latent space $\mathcal{Z}$ to find a latent representation $\tilde{z}$ that produces the desired classification outcome (Dombrowski et al., 2023). The iterative update rule in the latent space is given by:

$$
z^{(i+1)} = z^{(i)} + \lambda \frac{\partial (k \circ g)_t}{\partial z}\left(z^{(i)}\right),
\tag{8}
$$

with step size $\lambda$. This approach leverages the manifold structure encoded by $g$, ensuring that counterfactuals $\tilde{x} = g(\tilde{z})$ adhere to the data manifold $\mathcal{M}$. Consequently, the generated counterfactuals remain semantically meaningful, exhibit minimal deviation from the original samples, and achieve a classification flip.

Thus, diffeomorphic counterfactuals provide a theoretically sound foundation for generating meaningful sample pairs $(x, \tilde{x})$, which are subsequently utilized in the representation module to achieve robust alignment and accurate ITE estimation.

### 4.3. Contrastive Learning for Sample-level Alignment

**Counterfactual Contrastive Learning.** For each mini-batch of training samples, we generate a diffeomorphic counterfactual $\tilde{x}_i$ for each instance $x_i$. These factual–counterfactual pairs $(x_i, \tilde{x}_i)$ are treated as positive pairs, while instances from different samples form the candidate set for negative pairs. To maintain computational efficiency, a fixed number of negative samples are randomly selected from this set. This sampling strategy ensures sufficient contrastive information while minimizing computational overhead.

Each instance $x_i$ and its corresponding counterfactual $\tilde{x}_i$ are processed through an encoder network, and the resulting representations $\phi_E$ and $\tilde{\phi}_E$ are processed by the projection head network to yield $c_i$ and $\tilde{c}_i$, which facilitates the application of contrastive loss while enabling the encoder to retain more information in its learned features (Ho & Nvasconcelos, 2020). The contrastive loss is applied to learn invariant representations of factual–counterfactual pairs ensuring that $c_i$ and $\tilde{c}_i$ are closely aligned, effectively capturing the nuances of potential outcomes under different treatments. The contrastive loss function is defined as:

$$\mathcal{L}_c = -log \frac{exp\left(sim\left(c_i, \tilde{c}_i\right)/\tau_{temp}\right)}{\sum\limits_{j=1}^{\mathcal{N}_i} \mathbb{I}_{j \neq i} exp\left(sim\left(c_i, c_j\right)/\tau_{temp}\right)}, \quad (9)$$

where $sim(c_i, \tilde{c}_i) = \frac{c_i^T \tilde{c}_i}{\|c_i\|_2 \|\tilde{c}_i\|_2}$ represents the cosine similarity, $\mathcal{N}_i$ is the set of batch-size randomly selected negative samples, and $\tau_{temp}$ is the temperature coefficient.

The contrastive loss aligns directly with the theoretical bounds established in Lemma 4.1, which reflects the representation distances between counterfactuals and their factual counterparts, as well as between counterfactuals and factual samples from the opposite group. By minimizing these representation distances, the counterfactual loss $\epsilon_{CF}(h, \Phi)$ remains bounded, as detailed in Equation (4). The contrastive learning module achieves robust alignment, effectively reducing the error terms identified in Theorem 4.2:

$$disc(\Phi^t(x), \Phi^t(\tilde{x})) \quad \text{and} \quad disc(\Phi^t(\tilde{x}), \Phi^{1-t}(x)).$$

These alignment errors, which quantify the distances between factual–counterfactual pairs and between counterfactual samples and opposing group samples, directly impact the ITE estimation error. By minimizing $\mathcal{L}_c$, the learned representations align with the theoretical bounds, improving the reliability of ITE estimation.

**Prediction Head.** The learned balanced representations $\phi_E(x_i)$ are fed into two neural networks to predict potential outcomes for treatment ($t = 1$) and control ($t = 0$) (Assaad et al., 2021; Huang et al., 2023). The predicted outcomes are defined as $T_{\text{out}} = h(\phi_E(x_i), t_i = 1)$ and $C_{\text{out}} = h(\phi_E(x_i), t_i = 0)$, respectively. The predictive loss is given by:

$$\mathcal{L}_p = \frac{1}{n}\sum_{i=1}^{n} w_i \cdot \mathcal{L}\left(h\left(\phi_E\left(x_i\right), t_i\right), y_i\right), \quad (10)$$

where $w_i = \frac{t_i}{2u} + \frac{1-t_i}{2(1-u)}$, and $u = \frac{1}{n}\sum_{i=1}^{n} t_i$. The total loss $\mathcal{L}_t$ combines predictive loss, contrastive loss, and regularization:

$$\mathcal{L}_t = \mathcal{L}_p + \alpha\mathcal{L}_c + \beta\|W\|_2, \quad (11)$$

where $\alpha$ and $\beta$ are adjustable hyper-parameters that control the contributions of contrastive loss and regularization loss $\|\cdot\|_2$ on model weights $W$ to prevent overfitting.

We train our model by minimizing Equation (11) and provide the detailed counterfactual contrastive learning strategy in Algorithm 1 in the Appendix. This formulation ensures that the learned representations align factual and counterfactual samples while accurately predicting potential outcomes, ultimately reducing ITE estimation error.

### 4.4. Why FCCL Works

FCCL bridges theoretical guarantees and practical innovation to address key challenges in ITE estimation, particularly for sparse and boundary samples. By generating diffeomorphic counterfactuals that adhere to the data manifold, FCCL ensures that counterfactuals are both realistic and meaningful. This design reduces noise sensitivity and preserves semantic consistency, which is critical for obtaining reliable sample pairs. Furthermore, contrastive learning enforces distance constraints based on factual-counterfactual similarity and category consistency (as formalized in Lemma 4.1), mitigating distribution shifts through fine-grained sample-level alignment, and approximating the randomization achieved in RCTs. Together, these mechanisms reduce ITE estimation error, which is consistent with Theorem 4.2, and outperform traditional methods in scenarios with significant heterogeneity or limited data.

## 5. Experiments

To evaluate the effectiveness of FCCL, particularly its robustness under varying degrees of covariate dispersion, we employ six benchmark datasets: four synthetic, one semi-synthetic, and one real-world datasets. Comprehensive experiments across various datasets validate our claims, including performance evaluation, latent space analysis and boundary sample examination.

### 5.1. Datasets

**Synthetic**: We generate covariates from a multivariate normal distribution $\mathcal{N}\left(\mathbf{0}, \gamma \cdot \sigma^2 \cdot \left[\rho\mathbf{1}_p\mathbf{1}_p^\top + (1-\rho)\mathbf{I}_p\right]\right)$, where the scaling parameter $\gamma \in \{0.4, 0.7, 1.0, 1.2\}$ controls the degree of covariate dispersion, as shown in Figure 7. Each dataset consists of 800 units split into training (63%), validation (27%), and test (10%) sets. We generate 30 independent datasets for analysis. Detailed data generation steps are outlined in Appendix D.3.

**Semi-synthetic (IHDP)**: The Infant Health and Development Program (IHDP) dataset, modified by Hill (Hill, 2011), evaluates the effect of specialist home visits on children's cognitive scores. It includes 25 covariates describing infants and their mothers. Selection bias is simulated by systemati-

*Table 1.* Within-sample and out-of-sample mean and standard errors for the metrics (Lower is better) on IHDP dataset.

| Method | $\sqrt{\epsilon_{PEHE}^{within}}$ | $\epsilon_{ATE}^{within}$ | $\sqrt{\epsilon_{PEHE}^{out-of}}$ | $\epsilon_{ATE}^{out-of}$ |
|---|---|---|---|---|
| OLS-1 | 5.83(0.39) | 0.73(0.04) | 5.91(0.27) | 0.95(0.06) |
| OLS-2 | 2.42(0.16) | 0.14(0.02) | 2.55(0.16) | 0.31(0.02) |
| BART | 2.13(0.22) | 0.24(0.05) | 2.32(0.12) | 0.35(0.03) |
| KNN | 2.13(0.08) | 0.15(0.05) | 4.16(0.23) | 0.80(0.05) |
| DML | 2.45(0.12) | 0.20(0.05) | 2.60(0.14) | 0.33(0.05) |
| CFR-Wass | 0.71(0.04) | 0.27(0.03) | 0.83(0.08) | 0.28(0.03) |
| CFR-MMD | 0.77(0.05) | 0.25(0.04) | 0.92(0.09) | 0.28(0.04) |
| SITE | 0.84(0.05) | 0.30(0.04) | 0.98(0.07) | 0.32(0.05) |
| CITE | 0.59(0.06) | 0.11(0.02) | 0.67(0.14) | 0.14(0.02) |
| GANITE | 1.92(0.29) | 0.43(0.41) | 2.43(0.46) | 0.49(0.38) |
| ABCEI | 0.79(0.06) | 0.12(0.02) | 1.00(0.13) | 0.15(0.03) |
| CBRE | 0.59(0.06) | 0.11(0.02) | 0.66(0.07) | 0.13(0.02) |
| DIGNet | 0.60(0.04) | 0.15(0.02) | 0.67(0.07) | 0.16(0.02) |
| **FCCL** | **0.53(0.04)** | **0.09(0.01)** | **0.64(0.07)** | **0.12(0.02)** |

*Table 2.* Out-of-sample mean and standard errors for the rooted PEHE ($\sqrt{\epsilon_{PEHE}^{out-of}}$) on Synthetic datasets.

| Method | $\gamma = 0.4$ | $\gamma = 0.7$ | $\gamma = 1.0$ | $\gamma = 1.2$ |
|---|---|---|---|---|
| OLS-1 | 8.41(0.84) | 10.85(1.34) | 13.00(1.68) | 14.32(1.55) |
| OLS-2 | 5.94(0.60) | 7.64(0.96) | 9.13(1.18) | 10.07(1.11) |
| BART | 3.40(0.50) | 4.17(0.60) | 4.86(0.61) | 6.14(0.58) |
| KNN | 5.50(0.62) | 7.26(0.75) | 9.08(1.27) | 10.27(1.07) |
| DML | 3.96(0.54) | 5.30(0.62) | 6.08(0.80) | 8.46(0.77) |
| CFR-Wass | **2.48(0.06)** | 3.60(0.09) | 4.72(0.14) | 5.37(0.14) |
| CFR-MMD | 2.54(0.06) | 3.62(0.09) | 4.74(0.14) | 5.41(0.14) |
| SITE | 2.69(0.13) | 4.25(0.23) | 6.01(0.38) | 6.21(0.17) |
| CITE | 2.71(0.07) | 3.70(0.10) | 4.74(0.14) | 5.41(0.14) |
| GANITE | 4.69(0.06) | 6.16(0.07) | 7.33(0.07) | 8.11(0.08) |
| ABCEI | 2.75(0.06) | 3.57(0.09) | 4.73(0.13) | 5.19(0.12) |
| CBRE | 2.93(0.05) | 3.85(0.08) | 5.02(0.12) | 5.73(0.13) |
| DIGNet | 3.18(0.11) | 3.97(0.13) | 5.10(0.17) | 5.81(0.15) |
| **FCCL** | 2.58(0.06) | **3.50(0.09)** | **4.49(0.13)** | **5.12(0.12)** |

cally excluding a subset of treated samples, resulting in 747 instances (139 treated and 608 control). We use the same 100 datasets, following the standard practice in the field.

**Real-world (Jobs)**: This dataset examines the causal effect of job training (treatment) on income and employment status. It is reformulated as a binary classification task to predict unemployment (Dehejia & Wahba, 2002). This dataset contains 297 treated samples and 2,915 control samples. For consistency with prior work, we average over 10 experiments. The train, validation, and test splits are set to 56%, 24%, and 20%, respectively.

### 5.2. Metrics

For datasets with known true treatment effects, we evaluate using the rooted *Precision in Estimation of Heterogeneous Effect* $\sqrt{\epsilon_{PEHE}}$ and the absolute error of *Average Treatment Effect* $\epsilon_{ATE}$. For the Jobs, where ground-truth counterfactuals are unavailable, we adopt the *policy risk* $\mathcal{R}_{pol}(\pi_{\hat{f}})$ and the bias of *Average Treatment Effect on the Treated* prediction $\epsilon_{ATT}$. Detailed formulas are provided in Appendix C.

### 5.3. Baseline Approaches

We compare our FCCL against thirteen baselines, categorized as traditional and deep learning methods.

**Traditional Methods:** Ordinary least square (**OLS-1**), which uses treatment as a covariate to predict outcomes; (**OLS-2**), which predicts outcomes separately for each group; Bayesian additive regression trees (**BART**) leveraging a sum-of-trees structure; K-nearest neighbor (**KNN**) matches samples using $k$-nearest neighbors; Debiased machine learning (**DML**), handling confounding bias through orthogonal residual regression. **Deep learning Methods: CFR-Wass** (Shalit et al., 2017) and **CFR-MMD** (Shalit

et al., 2017) are two methods using the Wasserstein and MMD metric for counterfactual regression, respectively; **SITE** (Yao et al., 2018), which preserves local similarity in sample representations; **CITE** (Li & Yao, 2022) learns representation based on propensity score; **GANITE** (Yoon et al., 2018) implicitly learns counterfactual distribution using GANs; **ABCEI** (Du et al., 2021) balances distributions using adversarial learning; **CBRE** (Zhou et al., 2022) constructs an information loop during adversarial training to minimize information loss; **DIGNet** (Huang et al., 2024) utilizes individual propensity confusion and group distance minimization to learn the balanced representation.

### 5.4. Experimental Results

We evaluate FCCL's performance on multiple metrics across various datasets, focusing on robustness under varying covariate dispersion. Additionally, we conduct latent space analysis and examine boundary sample behavior to provide deeper insights into FCCL's effectiveness. Further results, including sensitivity analysis, are presented in Appendix D.

**Performance Evaluation:** We compare our FCCL against the baseline methods on IHDP, Jobs and Synthetic datasets, with results presented in Table 1 and Table 2, and additional results are provided in the Appendix. Key findings include: (1) FCCL outperforms all baseline methods across various datasets. Notably, on the IHDP dataset, FCCL achieves the lowest $\sqrt{\epsilon_{PEHE}}$ and $\epsilon_{ATE}$ values, reducing $\epsilon_{ATE}^{\text{out-of}}$ by 57.1% compared to CFR. (2) Methods leveraging contrastive learning (e.g., CITE, FCCL) consistently outperform CFR-based methods, highlighting their strength in aligning representations under diverse treatments. (3) FCCL shows resilience to covariate dispersion, as it maintains robust performance as the covariate dispersion ($\gamma$) increases from 0.4 to 1.2, with $\sqrt{\epsilon_{PEHE}^{out-of}}$ increasing from 2.58 to 5.12. This increase is 12.1% lower compared to CFR-Wass.

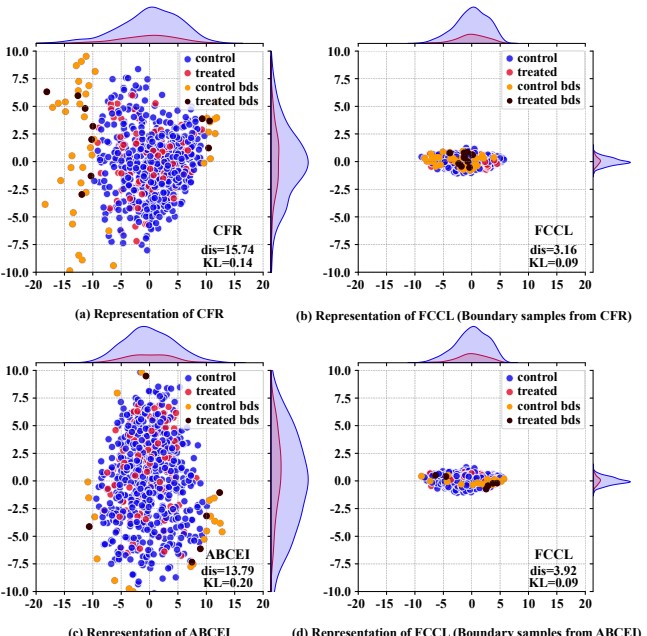

*Figure 3.* Latent Distributions of two representative methods, **CFR** and **ABCEI**, alongside our **FCCL** on IHDP dataset. The dots in black and yellow represent the boundary samples from the treatment and control groups, respectively. The metric *dis* quantifies the average distance between boundary samples and corresponding class centers, reflecting sample heterogeneity. The metric *KL* divergence is used to quantify the difference between treatment and control distributions.

**Latent Space Representation and Boundary Sample Analysis:** Figure 3 visualizes the latent space representation for CFR (covariate balance-based), ABCEI (adversarial training-based), and our proposed FCCL method (contrastive learning-based). FCCL generates well-balanced and compact representations and achieves lower KL divergence than CFR and ABCEI, significantly reducing distribution discrepancies between treatment and control groups through robust sample-level alignment. Additionally, boundary and heterogeneous samples limit the model's generalizability, especially in scenarios with substantial individual heterogeneity. A smaller "dis" value indicates that dispersion of samples in the latent representation space is reduced, which shows that there are fewer boundary samples, enabling better model fitting and exhibiting lower ITE estimation bias. Motivated by this, FCCL focuses on boundary samples—defined as the 30 farthest samples from their class centers—and achieves: (1) reduced dispersion: as FCCL aligns boundary samples closer to their class centers, outperforming CFR ($dis = 15.74$) and ABCEI ($dis = 13.79$); and (2) improved ITE estimation: analyzing ITE estimation bias (Tables 4 and 5 in Appendix D) shows that FCCL achieves lower estimation bias, for boundary samples compared to baselines.

*Table 3.* ITE estimation errors with different counterfactual generation strategies on IHDP dataset.

| Method | $\sqrt{\epsilon_{PEHE}^{within}}$ | $\epsilon_{ATE}^{within}$ | $\sqrt{\epsilon_{PEHE}^{out-of}}$ | $\epsilon_{ATE}^{out-of}$ |
|---|---|---|---|---|
| grad asc in $\mathcal{X}$ | 0.60(0.06) | 0.11(0.02) | 0.87(0.19) | 0.15(0.04) |
| GAN | 0.57(0.05) | 0.10(0.01) | 0.84(0.17) | 0.14(0.03) |
| diffusion | 0.54(0.05) | 0.09(0.01) | 0.72(0.10) | 0.12(0.03) |
| **FCCL** | **0.53(0.04)** | **0.09(0.01)** | **0.64(0.07)** | **0.12(0.02)** |

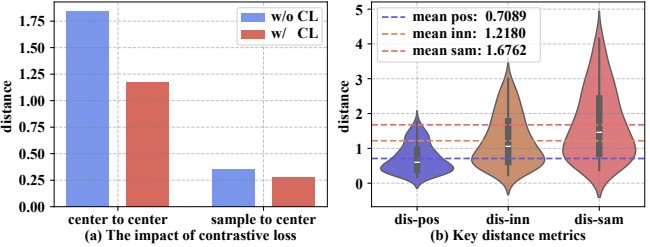

*Figure 4.* (a) The impact of contrastive loss (CL) on distribution balance, comparing distances between class centroids (center-to-center) and between samples and the center of the opposite class (sample-to-center) with and without CL. (b) Key distance metrics in the representation space: factual-counterfactual (dis-pos), counterfactual-opposite factual (dis-inn), and factual-factual (dis-sam), shown with mean values for each metric. Contrastive learning reduces distances, enhancing representation alignment and mitigating distribution discrepancies.

## 5.5. Components Analysis

This section explores the contribution of two key components of FCCL—counterfactual generation and contrastive learning—toward improving ITE estimation performance.

**Effectiveness of Counterfactual Generation:** To validate the efficacy of the flow-based counterfactual generation, we compare it with alternative strategies, including gradient ascent in the $\mathcal{X}$-space, GAN-based generation (Goodfellow et al., 2020), and diffusion-based generation (Kotelnikov et al., 2023). As shown in Table 3, FCCL significantly outperforms these methods, achieving a 26.4% reduction in $\sqrt{\epsilon_{PEHE}^{out-of}}$ compared to the gradient ascent in the $\mathcal{X}$-space. While the diffusion model achieves performance comparable to FCCL, it yields slightly inferior result on $\sqrt{\epsilon_{PEHE}^{out-of}}$. This may be due to the noise-driven generation mechanism of diffusion models, which causes counterfactual to deviate from the sample semantic space, especially in the out-sample cases. Unlike other approaches, which introduce noise and fail to preserve the inherent structure of the data, FCCL generates diffeomorphic counterfactuals that remain consistent with the data manifold that represents the intrinsic structure of the data. This structural consistency enhances causal effect estimation, making FCCL more reliable for individual-level treatment effect predictions.

**Impact of Contrastive Learning:** Contrastive learning plays a critical role in FCCL's robust performance by aligning representations of treated and control groups. As illustrated in Figure 4, contrastive loss reduces the class-center distances (center-to-center) and optimizes the distances between: (1) *counterfactual and corresponding factual samples*: where the mean positive distance (dis-pos) is 0.7089; (2) *counterfactual and opposite-class factual samples*, where the mean inner-class distance (dis-inn) is 1.2180. This alignment effectively balances the treated and control distributions, achieving sample-level correspondence and mitigating distribution discrepancies. By ensuring category consistency and factual-counterfactual similarity, FCCL achieves superior representation balance, as theoretically justified in Section 4.4.

## 6. Conclusion

In this paper, we proposed FCCL, a robust method for ITE estimation, grounded in our derived ITE estimation error bound. FCCL innovatively integrated diffeomorphic counterfactual generation and contrastive learning to address distribution shifts between treated and control groups through fine-grained, sample-level alignment. Comprehensive experiments across various datasets, including performance evaluation, latent space analysis and boundary sample examination, have demonstrated that FCCL achieves more accurate and robust ITE estimation, particularly in scenarios with significant individual heterogeneity. For future directions, we aim to extend our framework to accommodate time-dependent outcome variables.

## Impact Statement

FCCL represents a significant advancement in causal effect estimation by providing a robust and accurate method capable of handling complex scenarios with substantial individual heterogeneity. Its ability to achieve precise alignment at the sample level makes it particularly suited for applications in personalized decision-making. Potential domains of impact include digital marketing for tailoring customer strategies, social sciences for policy evaluation, and healthcare for personalized treatment planning.

## Acknowledgements

This work is supported by the National Natural Science Foundation of China (U24A20323, 62376145), the Science and Technology Innovation Talent Team of Shanxi Province (202204051002016), and the Key Technologies Program of Taihang Laboratory in Shanxi Province (THYF-JSZX-24010700).

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

# A. Proof.

**Definition A.1. Diffeomorphic counterfactual.** The diffeomorphic counterfactual is defined as minimal modifications on the data manifold of the input reversing the classification decision $k(x)$, represented as:

$$\tilde{x} = \arg \min_{\tilde{x} \in \mathcal{M}} dist(\tilde{x}, x) \; s.t. \; k(\tilde{x}) \neq k(x),$$

where $\mathcal{M}$ is the data manifold, $dist(\tilde{x}, x)$ denotes the distance between $\tilde{x}$ and $x$. The classifier $k : \mathcal{X} \to \{0, 1\}$ assigns an input $x \in \mathcal{X}$ to a class $t \in \{0, 1\}$, and $k(\tilde{x}) \neq k(x)$ indicates a change in the classifier's prediction outcome.

**Definition A.2.** Let $p^{t=1}(x) := p(x|t = 1)$, $p^{t=0}(x) := p(x|t = 0)$ denote the factual treatment and control distributions, respectively, and $p^{t=1}(\tilde{x}) := p(\tilde{x}|t = 1)$, $p^{t=0}(\tilde{x}) := p(\tilde{x}|t = 0)$ denote respectively the counterfactual treatment and control distributions.

**Definition A.3.** For a representation function $\Phi : \mathcal{X} \to \mathcal{R}$, and for a distribution $p$ defined over $\mathcal{X}$, let $p_\Phi$ be the distribution induced by $\Phi$ over $\mathcal{R}$. Define $p_\Phi^{t=1}(r) := p_\Phi(r|t = 1)$ and $p_\Phi^{t=0}(r) := p_\Phi(r|t = 0)$ to be the factual treatment and control distributions induced over $\mathcal{R}$, $p_\Phi^{t=1}(\tilde{r}) := p_\Phi(\tilde{r}|t = 1)$ and $p_\Phi^{t=0}(\tilde{r}) := p_\Phi(\tilde{r}|t = 0)$, to be the counterfactual treatment and control distributions induced over $\mathcal{R}$.

**Lemma A.4.**

$$\epsilon_F(h, \Phi) = u \cdot \epsilon_F^{t=1}(h, \Phi) + (1 - u) \cdot \epsilon_F^{t=0}(h, \Phi),$$
$$\epsilon_{CF}(h, \Phi) = (1 - u) \cdot \epsilon_{CF}^{t=1}(h, \Phi) + u \cdot \epsilon_{CF}^{t=0}(h, \Phi).$$

Define $u := p(t = 1)$ as the treatment proportion. The proof follows directly by noting that $p(x, t) = u \cdot p^{t=1}(x) + (1 - u) \cdot p^{t=0}(x)$ (Shalit et al., 2017).

**Definition A.5.** Let $L : \mathcal{Y} \times \mathcal{Y} \to \mathbb{R}_+$ be the absolute loss or squared loss, $l_{h,\Phi}(x, t)$ denote the expected loss for the unit-treatment pair $(x, t)$, with representation function $\Phi : \mathcal{X} \to \mathcal{R}$ and hypothesis $h : \mathcal{R} \times \{0, 1\} \to \mathcal{Y}$, which can be mathematically represented as: $l_{h,\Phi}(x, t) = \int_{\mathcal{Y}} L(y^t, h(\Phi(x), t)) p(y^t|x) dy^t$. The expected factual and counterfactual losses of $h$ and $\Phi$ are:

$$\epsilon_F(h, \Phi) = \int_{\mathcal{X} \times \{0,1\}} l_{h,\Phi}(x, t) p(x, t) dx dt,$$
$$\epsilon_{CF}(h, \Phi) = \int_{\mathcal{X} \times \{0,1\}} l_{h,\Phi}(x, t) p(x, 1 - t) dx dt.$$

**Definition A.6.** The expected factual treated and control losses are:

$$\epsilon_F^{t=1}(h, \Phi) = \int_{\mathcal{X}} l_{h,\Phi}(x, 1) p^{t=1}(x) dx,$$
$$\epsilon_F^{t=0}(h, \Phi) = \int_{\mathcal{X}} l_{h,\Phi}(x, 0) p^{t=0}(x) dx.$$

**Definition A.7.** The expected counterfactual treated and control losses are:

$$\epsilon_{CF}^{t=1}(h, \Phi) = \int_{\mathcal{X}} l_{h,\Phi}(x, 1) p^{t=0}(x) dx,$$
$$\epsilon_{CF}^{t=0}(h, \Phi) = \int_{\mathcal{X}} l_{h,\Phi}(x, 0) p^{t=1}(x) dx.$$

**Lemma A.8.** *(Lemma 4.1. main text). Let $\Phi : \mathcal{X} \to \mathcal{R}$ be an invertible representation function with inverse $\Psi$ and $h : \mathcal{R} \times \{0, 1\} \to \mathcal{Y}$ a hypothesis function. Define $u := p(t = 1)$ as the treatment proportion. For any treatment indicator $t \in \{0, 1\}$, let $disc(\Phi^t(x), \Phi^t(\tilde{x}))$ represent the distance between counterfactual and factual samples in the representation space. Then, the counterfactual loss $\epsilon_{CF}(h, \Phi)$ is bounded by:*

$$\epsilon_{CF}(h, \Phi) \leq (1 - u) \cdot \epsilon_F^{t=1}(h, \Phi) + u \cdot \epsilon_F^{t=0}(h, \Phi) +$$
$$disc(\Phi^t(x), \Phi^t(\tilde{x})) + disc(\Phi^t(\tilde{x}), \Phi^{1-t}(x)).$$

*Proof.*

$$\epsilon_{CF}(h, \Phi) - \left[(1 - u) \cdot \epsilon_F^{t=1}(h, \Phi) + u \cdot \epsilon_F^{t=0}(h, \Phi)\right] = \tag{12}$$

$$\left[(1 - u) \cdot \epsilon_{CF}^{t=1}(h, \Phi) + u \cdot \epsilon_{CF}^{t=0}(h, \Phi)\right] -$$

$$\left[(1 - u) \cdot \epsilon_F^{t=1}(h, \Phi) + u \cdot \epsilon_F^{t=0}(h, \Phi)\right] =$$

$$(1 - u) \cdot [\epsilon_{CF}^{t=1}(h, \Phi) - \epsilon_F^{t=1}(h, \Phi)]+$$

$$u \cdot [\epsilon_{CF}^{t=0}(h, \Phi) - \epsilon_F^{t=0}(h, \Phi)] = \tag{13}$$

$$(1 - u) \int_{\mathcal{X}} l_{h,\Phi}(x, 1)(p^{t=0}(x) - p^{t=1}(x))dx+$$

$$u \int_{\mathcal{X}} l_{h,\Phi}(x, 0)(p^{t=1}(x) - p^{t=0}(x))dx =$$

$$(1 - u) \int_{\mathcal{X}} l_{h,\Phi}(x, 1)(p^{t=0}(x) + p^{t=0}(\tilde{x}) - p^{t=0}(\tilde{x}) - p^{t=1}(x))dx+$$

$$u \int_{\mathcal{X}} l_{h,\Phi}(x, 0)(p^{t=1}(x) + p^{t=1}(\tilde{x}) - p^{t=1}(\tilde{x}) - p^{t=0}(x))dx =$$

$$(1 - u) \int_{\mathcal{X}} l_{h,\Phi}(x, 1)(p^{t=0}(x) - p^{t=0}(\tilde{x}))dx+$$

$$(1 - u) \int_{\mathcal{X}} l_{h,\Phi}(x, 1)(p^{t=0}(\tilde{x}) - p^{t=1}(x))dx+$$

$$u \int_{\mathcal{X}} l_{h,\Phi}(x, 0)(p^{t=1}(x) - p^{t=1}(\tilde{x}))dx+$$

$$u \int_{\mathcal{X}} l_{h,\Phi}(x, 0)(p^{t=1}(\tilde{x}) - p^{t=0}(x))dx = \tag{14}$$

$$(1 - u) \left[\sum_i l_{h,\Phi}(x_i, 1)(p^{t=0}(x_i) - p^{t=0}(\tilde{x}_i))\right] +$$

$$(1 - u) \left[\sum_i l_{h,\Phi}(x_i, 1)(p^{t=0}(\tilde{x}_i)) - p^{t=1}(x_i))\right] +$$

$$u \left[\sum_i l_{h,\Phi}(x_i, 0)(p^{t=1}(x_i) - p^{t=1}(\tilde{x}_i))\right] +$$

$$u \left[\sum_i l_{h,\Phi}(x_i, 0)(p^{t=1}(\tilde{x}_i)) - p^{t=0}(x_i))\right] \leq \tag{15}$$

$$(1 - u) \sum_i a_i \left|(p^{t=0}(x_i) - p^{t=0}(\tilde{x}_i))\right| +$$

$$(1 - u) \sum_i a_i \left|(p^{t=0}(\tilde{x}_i)) - p^{t=1}(x_i))\right| +$$

$$u \sum_i b_i \left|(p^{t=1}(x_i) - p^{t=1}(\tilde{x}_i))\right| +$$

$$u \sum_i b_i \left|(p^{t=1}(\tilde{x}_i)) - p^{t=0}(x_i))\right| =$$

$$(1 - u) \sum_i a_i \left|(p_\Phi^{t=0}(r_i) - p_\Phi^{t=0}(\tilde{r}_i))\right| +$$

$$(1 - u) \sum_i a_i \left|(p_\Phi^{t=0}(\tilde{r}_i)) - p_\Phi^{t=1}(r_i))\right| +$$

$$u \sum_i b_i \left|(p_\Phi^{t=1}(r_i) - p_\Phi^{t=1}(\tilde{r}_i))\right| +$$

$$u \sum_i b_i \left| (p_\Phi^{t=1}(\tilde{r}_i)) - p_\Phi^{t=0}(r_i)) \right| \approx$$

$$(1-u) \sum_i a_i \left| (\Phi^{t=0}(x_i) - \Phi^{t=0}(\tilde{x}_i)) \right| +$$

$$(1-u) \sum_i a_i \left| (\Phi^{t=0}(\tilde{x}_i)) - \Phi^{t=1}(x_i)) \right| +$$

$$u \sum_i b_i \left| (\Phi^{t=1}(x_i) - \Phi^{t=1}(\tilde{x}_i)) \right| +$$

$$u \sum_i b_i \left| (\Phi^{t=1}(\tilde{x}_i)) - \Phi^{t=0}(x_i)) \right| = \tag{16}$$

$$(1-u) \sum_i a_i \sqrt{2 - 2\Phi^{t=0}(x_i)^T \Phi^{t=0}(\tilde{x}_i)} +$$

$$(1-u) \sum_i a_i \sqrt{2 - 2\Phi^{t=0}(\tilde{x}_i)^T \Phi^{t=1}(x_i)} +$$

$$u \sum_i b_i \sqrt{2 - 2\Phi^{t=1}(x_i)^T \Phi^{t=1}(\tilde{x}_i)} +$$

$$u \sum_i b_i \sqrt{2 - 2\Phi^{t=1}(\tilde{x}_i)^T \Phi^{t=0}(x_i)} =$$

$$disc(\Phi^t(x), \Phi^t(\tilde{x})) + disc(\Phi^t(\tilde{x}), \Phi^{1-t}(x)).$$

$$\square$$

Equation (12) is by Lemma A.8. Equation (13) is by Definition A.6 and Definition A.7, the expected factual treated and control losses and the expected counterfactual treated and control losses. Equation (14) is based on the definition of integral formulation. The inequality (15) is by the absolute value inequality, which states that $a - b \leq |a - b|$. Equation (16) is by the law of Cosines and based on the assumption that $\Phi(x)$ is a unit vector.

The counterfactual loss $\epsilon_{CF}(h, \Phi)$ is bounded by a combination of factual prediction errors and the distances between factual–counterfactual pairs $disc(\Phi^t(x), \Phi^t(\tilde{x}))$ and between counterfactuals and opposing group factuals $disc(\Phi^t(\tilde{x}), \Phi^{1-t}(x))$. This Lemma highlights the importance of minimizing these representation distances to reduce counterfactual loss.

**Definition A.9.** The individual treatment effect:

$$\tau(x) := \mathbb{E}\left[y^1 - y^0 | x\right].$$

**Definition A.10.** Let $f : \mathcal{X} \times \{0, 1\} \to \mathcal{Y}$ by an hypothesis. The estimated individual treatment effect:

$$\hat{\tau}_f(x) = f(x, 1) - f(x, 0).$$

**Definition A.11.** The expected Precision in Estimation of Heterogeneous Effect loss of $f$:

$$\epsilon_{PEHE}(f) = \int_{\mathcal{X}} (\hat{\tau}_f(x) - \tau(x))^2 p(x) dx.$$

Following CFR (Shalit et al., 2017), we define $m_t(x) := \mathbb{E}[y^t|x]$, the expected variance of $y^t$ with respect to a distribution $p(x, t)$ is: $\sigma_{y^t}^2(p(x, t)) = \int_{\mathcal{X} \times \mathcal{Y}} (y^t - m_t(x))^2 p(y^t|x)p(x, t)dy^t dx$, and $\sigma_Y^2 = \min\left\{\sigma_{y^t}^2(p(x, t)), \sigma_{y^t}^2(p(x, 1-t))\right\}$. For any function $f : \mathcal{X} \times \{0, 1\} \to \mathcal{Y}$, and distribution $p(x, t)$ over $\mathcal{X} \times \{0, 1\}$, recall that $\int_{\mathcal{X}} (f(x, t) - m_t(x))^2 p(x, t)dxdt = \epsilon_F(f) - \sigma_{y^t}^2(p(x, t)), \int_{\mathcal{X}} (f(x, t) - m_t(x))^2 p(x, 1-t)dxdt = \epsilon_{CF}(f) - \sigma_{y^t}^2(p(x, 1-t))$. Next, we provide an upper bound for the expected Precision in Estimation of Heterogeneous Effect.

**Theorem A.12.** (Theorem 4.2. main text). Let $\epsilon_{PEHE}(h, \Phi)$ be the estimation error with representation function $\Phi : \mathcal{X} \to \mathcal{R}$ with $\Psi$ being its inverse and hypothesis $h : \mathcal{R} \times \{0, 1\} \to \mathcal{Y}$. For any treatment indicator $t \in \{0, 1\}$, let $disc(\Phi^t(x), \Phi^t(\tilde{x}))$ be the representation distance between counterfactual and factual samples. The error is bounded by:

$$\epsilon_{PEHE}(h, \Phi) \leq$$
$$2(\epsilon_F^{t=1}(h, \Phi) + \epsilon_F^{t=0}(h, \Phi) - 2\sigma_Y^2) +$$
$$2(disc(\Phi^t(x), \Phi^t(\tilde{x})) + disc(\Phi^t(\tilde{x}), \Phi^{1-t}(x))).$$

*Proof. Recall that we denote $\epsilon_{PEHE}(f) = \epsilon_{PEHE}(h, \Phi)$ for $f(x, t) = h(\Phi(x), t)$ .*

$$\epsilon_{PEHE}(f) =$$
$$= \int_{\mathcal{X}} ((f(x, 1) - f(x, 0)) - (m_1(x) - m_0(x)))^2 \, p(x) dx =$$
$$= \int_{\mathcal{X}} ((f(x, 1) - m_1(x)) + (m_0(x) - f(x, 0)))^2 \, p(x) dx \leq \tag{17}$$
$$2 \int_{\mathcal{X}} \left( (f(x, 1) - m_1(x))^2 + (m_0(x) - f(x, 0))^2 \right) p(x) dx = \tag{18}$$
$$2 \int_{\mathcal{X}} (f(x, 1) - m_1(x))^2 \, p(x, t = 1) dx +$$
$$2 \int_{\mathcal{X}} (m_0(x) - f(x, 0))^2 \, p(x, t = 0) dx +$$
$$2 \int_{\mathcal{X}} (f(x, 1) - m_1(x))^2 \, p(x, t = 0) dx +$$
$$2 \int_{\mathcal{X}} (m_0(x) - f(x, 0))^2 \, p(x, t = 1) dx =$$
$$2 \int_{\mathcal{X}} (f(x, t) - m_t(x))^2 \, p(x, t) dx dt +$$
$$2 \int_{\mathcal{X}} (f(x, t) - m_t(x))^2 \, p(x, 1 - t) dx dt \leq$$
$$2 \left( \epsilon_F - \sigma_Y^2 \right) + 2 \left( \epsilon_{CF} - \sigma_Y^2 \right).$$

where Equation (17) is because $(x + y)^2 \leq 2(x^2 + y^2)$, Equation (18) is because $p(x) = p(x, t = 0) + p(x, t = 1)$.

Then, we can prove Theorem A.12 based on Lemma A.4 and Lemma A.8:

$$\epsilon_{PEHE}(h, \Phi) \leq$$
$$2(\epsilon_F(h, \Phi) + \epsilon_{CF}(h, \Phi) - 2\sigma_Y^2) \leq$$
$$2(\epsilon_F^{t=1}(h, \Phi) + \epsilon_F^{t=0}(h, \Phi) - 2\sigma_Y^2) +$$
$$2(disc(\Phi^t(x), \Phi^t(\tilde{x})) + disc(\Phi^t(\tilde{x}), \Phi^{1-t}(x))).$$

$\square$

The estimation error $\epsilon_{PEHE}(h, \Phi)$ is upper bounded by two terms: the standard generalization error of factual prediction $\epsilon_F$ and the distance constraints in the representation space. These distance constraints represent two types of alignment: between counterfactual and factual samples $\Phi^t(x)^T \Phi^t(\tilde{x})$, and between counterfactual and opposite group factual samples $\Phi^t(\tilde{x})^T \Phi^{1-t}(x)$. By using contrastive loss to minimize these distances, sample-wise alignment is achieved, which helps reduce the expected ITE estimation error by learning consist representations that effectively capture the characteristics of potential outcomes under different treatments.

# B. Additional Technical Details

Based on reliable sample pairs $(x, \tilde{x})$, we leverage contrastive learning to effectively learn consistent representations for individuals across different treatments, further obtaining balanced representations through fine-grained, sample-level alignment. Additionally, we use two separate neural networks to estimate potential outcomes under varying treatments. The detailed counterfactual contrastive learning strategy is provided in Algorithm 1.

---

**Algorithm 1** Counterfactual Contrastive Learning

---

1: **Input:** Original dataset $\mathcal{D} = (x, t, y)$, batch size $N$, normalizing flow $g$, classifier $k$, step size $\lambda$, threshold $\Lambda$, temperature $\tau_{temp}$, Encoder network $Enc$, Projection head $Pro$, prediction head $h$, hyper-parameters $\alpha, \beta \geq 0$.
2: **Output:** Predicted factual $\hat{y}_f$, counterfactual $\hat{y}_{cf}$.
3: Split $\mathcal{D}$ into training set $\mathcal{D}_{\text{train}}$ and validation set $\mathcal{D}_{\text{valid}}$.
4: # Stage 1: Generate counterfactuals for each sample
5: **for** each mini-batch in $\mathcal{D}_{\text{train}}$ **do**
6:     $z \leftarrow g^{-1}(x)$
7:     **for** $i = 1$ to $N$ **do**
8:         $\nabla_z \leftarrow \frac{\partial (k \circ g)_t}{\partial z}$
9:         $\tilde{z} \leftarrow \text{optimizer.step}(\lambda, \nabla_z)$
10:         **if** $k(g(\tilde{z}))_t > \Lambda$ **then**
11:             **return** $g(\tilde{z})$
12:         **end if**
13:     **end for**
14:     Obtain counterfactuals $\{\tilde{x}_i\}_{i=1}^N$ from $\tilde{\mathcal{D}}$.
15: **end for**
16: # Stage 2: Train encoder using contrastive loss with generated counterfactuals
17: **repeat**
18:     Compute embeddings $c_i = Pro(Enc(x_i))$, $\tilde{c}_i = Pro(Enc(\tilde{x}_i))$
19:     Calculate contrastive loss $\mathcal{L}_c$ using Equation (9).
20:     Calculate predictive loss $\mathcal{L}_p$ using Equation (10), obtain predicted factual $\hat{y}_f$ and counterfactual $\hat{y}_{cf}$.
21:     Compute total loss $\mathcal{L}_t = \mathcal{L}_p + \alpha\mathcal{L}_c + \beta\|W\|_2$.
22:     Update Encoder network and Projection head.
23:     Terminate if converged on validation set $\mathcal{D}_{\text{valid}}$.
24: **until** convergence

---

# C. Metrics

For datasets with known true treatment effects, we adopt two commonly evaluation metrics(Alaa & Van Der Schaar, 2017; Cheng et al., 2022; Wu et al., 2022), namely the rooted Precision in Estimation of Heterogeneous Effect ($\sqrt{\epsilon_{PEHE}}$) and the absolute error of Average Treatment Effect ($\epsilon_{ATE}$) defined as:

$$\sqrt{\epsilon_{PEHE}} = \sqrt{\frac{1}{n} \sum_{i=1}^{n} (\tau(x_i) - \hat{\tau}(x_i))^2}, \tag{19}$$

$$\epsilon_{ATE} = \left| \hat{ATE} - ATE \right| = \frac{1}{n} \left| \sum_{i=1}^{n} (\tau_i - \hat{\tau}_i) \right|, \tag{20}$$

where $\tau_i$ refer to the ground truth treatment effect of a subject, $\hat{\tau}_i$ is the estimated treatment effect.

On Jobs dataset where the ground truth counterfactual outcomes are unavailable(Zhu et al., 2021; Reddy & Balasubramanian, 2024), we employ the policy risk $\mathcal{R}_{pol}(\pi_{\hat{\tau}})$ and the bias of Average Treatment Effect on the Treated prediction $\epsilon_{ATT}$ as evaluation metrics:

$$\mathcal{R}_{pol}(\pi_{\hat{\tau}}) = 1 - \left\{ p(\pi_{\hat{\tau}}(x) = 1) \cdot \mathbb{E}[y^1 | \pi_{\hat{\tau}}(x) = 1] \right.$$
$$\left. + p(\pi_{\hat{\tau}}(x) = 0) \cdot \mathbb{E}[y^0 | \pi_{\hat{\tau}}(x) = 0] \right\}, \tag{21}$$

$$\epsilon ATT = \left| \left| \frac{1}{|\mathcal{T}_1|} \sum_{i=1}^{|\mathcal{T}_1|} y_i^1 - \frac{1}{|\mathcal{T}_0|} \sum_{i=1}^{|\mathcal{T}_0|} y_i^0 \right| - \left| \frac{1}{|\mathcal{T}_1|} \sum_{i=1}^{|\mathcal{T}_1|} \left( \hat{y_i^1} - \hat{y_i^0} \right) \right| \right|, \tag{22}$$

where $\pi_{\hat{\tau}} : \mathcal{X} \to \{0, 1\}$ is a policy induced from an ITE estimator $\hat{\tau}(\cdot)$ with $\pi_{\hat{\tau}}(x) = 1$ if $\hat{\tau}(x) > 0$ and $\hat{\tau}(x) = 0$ otherwise. This measures the average regret when treating with the induced policy $\pi_{\hat{\tau}}$ and thus can serve as a proxy of the ITE estimation error. $|\mathcal{T}_1|$ and $|\mathcal{T}_0|$ are the number of the units in the treatment and the control groups, respectively.

## D. Additional Experiments.

### D.1. Additional Experiments on IHDP dataset.

Existing covariate balance methods mitigate treatment selection bias across treatment and control groups by enforcing distributional balance, but fail to account for boundary samples, which are scarce and situated far from the main data distribution. These sparse boundary samples limit model's generalizability, which is particularly pronounced in scenarios with substantial individual heterogeneity. Focusing on and analyzing these samples can significantly enhance model robustness and effectively reduce estimation errors.

We first provide a formal definition of boundary samples. Let the feature matrix be $\mathcal{X} \in \mathbb{R}^{n \times d}$, where $n$ is the number of samples and $d$ is a $d$-dimensional feature vector (e.g., age, sex). Let $t \in \{0, 1\}$ denote the treatment assignment for each unit $i$, where $t_i = 1$ indicates the treated sample, and $t_i = 0$ indicates the sample belongs to the control group.

The treated group center $u_1$ and the control group center $u_0$ are calculated as the mean values within each group:

$$u_0 = \frac{1}{|\mathcal{T}_0|} \sum_{i \in \mathcal{T}_0} x_i, \quad \mathcal{T}_0 = \{i | t_i = 0\},$$

$$u_1 = \frac{1}{|\mathcal{T}_1|} \sum_{i \in \mathcal{T}_1} x_i, \quad \mathcal{T}_1 = \{i | t_i = 1\}.$$

For each sample $x_i \in \mathbb{R}^d$, we calculate the distances between $x_i$ and the treated group center, and the control group center respectively: $d_1(x_i) = \|x_i - u_1\|$ and $d_0(x_i) = \|x_i - u_0\|$. Taking the treated group as an example, we first sort all samples in $\mathcal{T}_1$ by their distance to the control group center $d_0(x)$ in descending order and select samples with large distances to exclude those in the overlapping region between treated and control groups. Among these candidates, we then select the top $k_1$ samples with the largest distance to their own group center $d_1(x)$ as treated group boundary samples. The value of $k_1$ is determined by the treated ratio $n_1 = \frac{|\mathcal{T}_1|}{|\mathcal{T}_0| + |\mathcal{T}_1|}$, such that $k_1 = n_1 \cdot k$, where $k$ is the total number of boundary samples.

As previously mentioned, existing deep learning-based causal inference methods can generally be grouped into two categories. The first category, exemplified by CFR, focuses on balancing the distributions of covariates between treated and control groups in the representation space to mitigate selection bias. The second category, represented by ABCEI, uses adversarial training to implicitly model the counterfactual distribution. To evaluate the robustness of different methods in estimating ITE, we identify 30 boundary samples corresponding to each of these two representative methods (CFR and ABCEI) and calculate the individual treatment effect estimation bias ($\epsilon_{ITE} = |ITE_{pred} - ITE_{true}|$) for each boundary sample. Meanwhile, we also calculate the ITE estimation bias for these same samples using our FCCL. The results are shown in Table 4 and Table 5.

We observe that our FCCL achieves significantly better performance than CFR and ABCEI. For example, compared to CFR, 22 out of 30 boundary samples achieve lower estimation errors, with an average error reduction of 81.83%. Similarly, our FCCL also demonstrates impressive improvements over ABCEI, with 23 out of 30 samples achieving significantly smaller ITE estimation errors. These results show that FCCL enables robust sample-level alignment, achieving more accurate and stable ITE estimation.

*Table 4.* The individual treatment effect estimation error of boundary samples for CFR-MMD and FCCL on IHDP dataset.

| Sample | $ITE_{true}$ | CFR-MMD | | FCCL | | Sample | $ITE_{true}$ | CFR-MMD | | FCCL | |
|---|---|---|---|---|---|---|---|---|---|---|---|
| | | $ITE_{pred}$ | $\|\epsilon_{ITE}\|$ | $ITE_{pred}$ | $\|\epsilon_{ITE}\|$ | | | $ITE_{pred}$ | $\|\epsilon_{ITE}\|$ | $ITE_{pred}$ | $\|\epsilon_{ITE}\|$ |
| 1 | 3.5205 | -0.7059 | 4.2264 | 3.7350 | **0.2144** | 16 | 3.8053 | 0.8315 | **2.9738** | -4.0517 | 7.8569 |
| 2 | 2.8730 | 5.9707 | 3.0977 | 2.8136 | **0.0594** | 17 | 2.7568 | -1.8170 | **4.5739** | -3.1549 | 5.9117 |
| 3 | 4.3918 | 4.3581 | **0.0337** | 4.3113 | 0.0805 | 18 | 0.9267 | 0.2564 | 0.6703 | 1.5344 | **0.6077** |
| 4 | 0.9640 | -5.1606 | 6.1246 | 1.6058 | **0.6418** | 19 | 3.9234 | 0.9879 | 2.9355 | 4.0710 | **0.1477** |
| 5 | -0.7087 | -2.8101 | 2.1014 | -0.4824 | **0.2263** | 20 | 2.8283 | -1.0772 | 3.9056 | 3.0010 | **0.1726** |
| 6 | 3.1767 | 2.8624 | **0.3143** | 3.6967 | 0.5200 | 21 | 4.4578 | 1.3300 | 3.1278 | 4.5571 | **0.0993** |
| 7 | 2.1111 | -0.6245 | **2.7356** | -2.1950 | 4.3061 | 22 | 3.4389 | 2.1831 | 1.2558 | 3.5603 | **0.1215** |
| 8 | 4.5069 | 1.4256 | 3.0813 | 4.6920 | **0.1850** | 23 | 4.4578 | -0.0649 | 4.5226 | 4.8985 | **0.4407** |
| 9 | 4.3352 | 1.7478 | 2.5874 | 4.5570 | **0.2218** | 24 | 4.3395 | 0.9931 | 3.3464 | 4.2455 | **0.0940** |
| 10 | 4.6119 | 2.6889 | 1.9231 | 4.8943 | **0.2824** | 25 | 4.5317 | 2.3423 | 2.1894 | 4.7399 | **0.2082** |
| 11 | 4.5069 | -3.3979 | **7.9049** | -4.7954 | 9.3024 | 26 | 0.3966 | -4.1701 | 4.5667 | 0.3130 | **0.0836** |
| 12 | 3.0626 | -0.6165 | 3.6791 | 3.5241 | **0.4615** | 27 | 4.5836 | -1.3938 | 5.9774 | 5.0885 | **0.5049** |
| 13 | 4.6119 | 1.5407 | 3.0712 | 4.9734 | **0.3615** | 28 | 1.9660 | -0.9089 | **2.8749** | -1.4851 | 3.4511 |
| 14 | 3.6437 | -3.7587 | 7.4024 | -3.7375 | **7.3812** | 29 | 4.5283 | 4.1638 | 0.3646 | 4.2638 | **0.2645** |
| 15 | 4.4007 | -2.5881 | **6.9888** | -4.4562 | 8.8569 | 30 | 4.1820 | 1.9005 | 2.2815 | 4.1756 | **0.0064** |

*Table 5.* The individual treatment effect estimation error of boundary samples for ABCEI and FCCL on IHDP dataset.

| Sample | $ITE_{true}$ | ABCEI | | FCCL | | Sample | $ITE_{true}$ | ABCEI | | FCCL | |
|---|---|---|---|---|---|---|---|---|---|---|---|
| | | $ITE_{pred}$ | $\|\epsilon_{ITE}\|$ | $ITE_{pred}$ | $\|\epsilon_{ITE}\|$ | | | $ITE_{pred}$ | $\|\epsilon_{ITE}\|$ | $ITE_{pred}$ | $\|\epsilon_{ITE}\|$ |
| 1 | 4.6441 | 3.9641 | 0.6800 | 4.6739 | **0.0298** | 16 | 4.6354 | 7.2090 | 2.5736 | 4.7878 | **0.1524** |
| 2 | 4.6577 | 4.6809 | **0.0232** | 4.7763 | 0.1186 | 17 | 1.7826 | 1.2860 | **0.4967** | 3.1989 | 1.4163 |
| 3 | 4.6627 | 5.2865 | 0.6238 | 4.9184 | **0.2558** | 18 | 1.7884 | 1.2142 | 0.5742 | 2.3505 | **0.5620** |
| 4 | -1.8670 | -3.3532 | 1.4862 | -2.5838 | **0.7168** | 19 | 3.4047 | 3.0164 | 0.3883 | 3.1130 | **0.2918** |
| 5 | 0.5711 | 0.5848 | **0.0137** | -0.8183 | 1.3895 | 20 | 4.6441 | 3.0385 | 1.6056 | 4.6846 | **0.0405** |
| 6 | 4.5688 | 5.4534 | 0.8846 | 4.7526 | **0.1838** | 21 | 4.6354 | -0.8909 | **5.5263** | -4.8816 | 9.5170 |
| 7 | 2.1111 | -1.7264 | **3.8375** | -2.3312 | 4.4423 | 22 | 4.6687 | 3.5674 | 1.1013 | 4.8389 | **0.1701** |
| 8 | 4.6701 | 3.0477 | 1.6225 | 4.8280 | **0.1579** | 23 | 4.6687 | 1.9664 | 2.7024 | 4.8932 | **0.2244** |
| 9 | 4.5480 | 3.3778 | 1.1702 | 4.6724 | **0.1244** | 24 | 4.6701 | -4.1481 | 8.8182 | -5.2006 | 9.8707 |
| 10 | 4.4591 | -1.0039 | 5.4630 | 4.7291 | **0.2701** | 25 | 1.1268 | -2.0455 | 3.1723 | 0.7695 | **0.3573** |
| 11 | 4.3369 | 3.8228 | 0.5141 | 4.5132 | **0.1763** | 26 | 3.9765 | 4.4113 | 0.4348 | 4.2919 | **0.3153** |
| 12 | 4.6577 | 5.4907 | 0.8329 | 4.7580 | **0.1003** | 27 | 4.6672 | 3.1701 | 1.4971 | 4.7203 | **0.0531** |
| 13 | 4.6701 | 2.9026 | 1.7676 | 4.9135 | **0.2434** | 28 | 4.6517 | -5.0671 | 9.7188 | -4.9836 | 9.6353 |
| 14 | 4.6627 | 1.9303 | 2.7324 | 4.9797 | **0.3170** | 29 | 4.4322 | 1.9372 | 2.4950 | 4.5838 | **0.1516** |
| 15 | 4.5855 | 2.2245 | 2.3610 | 4.8239 | **0.2384** | 30 | 4.6687 | -3.1260 | **7.7947** | -4.9109 | 9.5797 |

We perform sensitivity analysis focusing on three key parameters: the weight of contrastive loss $\alpha$, the weight of model complexity term $\beta$, and the temperature coefficient $\tau_{temp}$ in the contrastive loss. In particular, we set $\alpha \in \{0.1, 0.3, 0.5, 0.8\}$, $\beta \in \{1\text{e-}3, 1\text{e-}4, 1\text{e-}5, 1\text{e-}6\}$, and $\tau_{temp} \in \{0.1, 0.3, 0.5, 0.7, 0.9\}$. The results in Figure 5 show that the contrastive loss, as one of the core optimization objectives in our proposed FCCL, impacts the individual treatment effect estimation performance. We find that FCCL is generally robust to different settings of these parameters. However, fine-tuning these hyperparameters can still enhance ITE estimation performance.

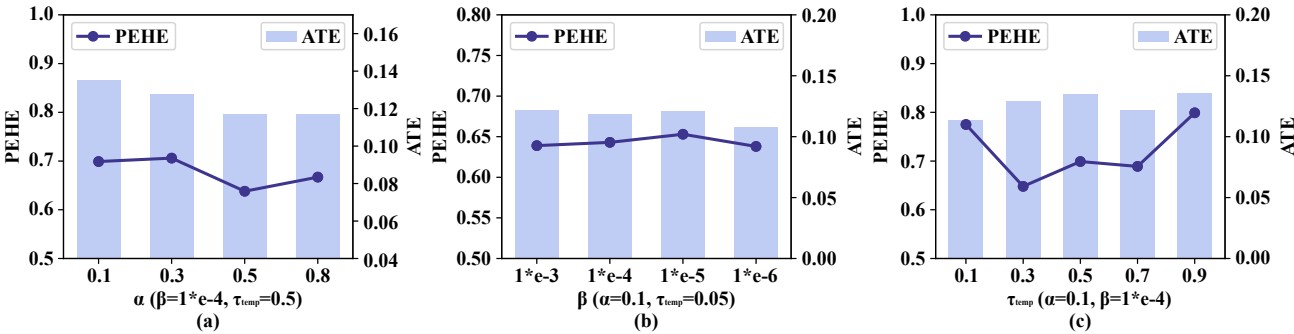

*Figure 5.* ITE estimation performance of our FCCL under different parameters on IHDP dataset.

### D.2. Additional Experiments on Jobs dataset.

On the Jobs dataset, although the performance of representation learning is less satisfactory compared to adversarial training to some extent, FCCL achieves comparable results to the baselines, particularly in the error metric $\epsilon_{ATT}$. The main reason we identified is the limited data size on the Jobs dataset, which is further evidenced by the observation that traditional methods often outperform deep learning on this dataset.

We also provide sensitivity analysis on Jobs dataset focusing on the weight of contrastive learning $\alpha$, the weight of model complexity term $\beta$, and the temperature coefficient $\tau_{temp}$. In particular, we vary the weight of the contrastive loss $\alpha \in \{0.2, 0.5, 0.8, 1.0, 1.2\}$ while fixing $\beta = 1\text{e-}4$ and $\tau_{temp} = 0.2$. Similarly, we explore the effect of the regularization weight $\beta \in \{1\text{e-}3, 1\text{e-}4, 1\text{e-}5, 1\text{e-}6\}$ with $\alpha = 1.0$ and $\tau_{temp} = 0.2$. For the temperature coefficient, we evaluate $\tau_{temp} \in \{0.1, 0.3, 0.5, 0.7, 0.9\}$ while keeping $\alpha = 1.0$ and $\beta = 1\text{e-}5$. From the results in Figure 6, we find our model is generally robust to different $\alpha$ settings and the best performance is achieved with $\alpha = 1.0$.

*Table 6.* Within-sample and out-of-sample policy risk and error on the average treatment effect on the treated (ATT) for the various models on Jobs dataset.

| Method | $R_{pol}^{within}$ | $\epsilon_{ATT}^{within}$ | $R_{pol}^{out-of}$ | $\epsilon_{ATT}^{out-of}$ |
|---|---|---|---|---|
| OLS-1 | 0.22(0.02) | 0.01(0.00) | 0.23(0.02) | 0.08(0.04) |
| OLS-2 | 0.21(0.01) | 0.01(0.01) | 0.24(0.03) | 0.08(0.03) |
| BART | 0.23(0.00) | 0.02(0.02) | 0.25(0.02) | 0.08(0.04) |
| KNN | 0.23(0.01) | 0.02(0.00) | 0.26(0.02) | 0.13(0.05) |
| DML | 0.23(0.01) | 0.02(0.02) | 0.24(0.02) | 0.10(0.03) |
| CFR-Wass | 0.23(0.01) | 0.06(0.02) | 0.26(0.02) | 0.10(0.04) |
| CFR-MMD | 0.22(0.00) | 0.07(0.03) | 0.27(0.01) | 0.12(0.05) |
| SITE | 0.23(0.01) | 0.05(0.02) | 0.25(0.02) | 0.10(0.04) |
| CITE | 0.23(0.00) | 0.10(0.03) | 0.26(0.02) | 0.13(0.05) |
| GANITE | **0.14(0.02)** | 0.27(0.74) | **0.15(0.01)** | 0.31(0.56) |
| ABCEI | 0.17(0.02) | 0.05(0.02) | 0.22(0.02) | 0.15(0.08) |
| CBRE | 0.30(0.00) | 0.08(0.03) | 0.31(0.00) | 0.11(0.03) |
| DIGNet | 0.23(0.01) | 0.07(0.04) | 0.26(0.01) | 0.13(0.04) |
| **FCCL** | 0.23(0.01) | **0.05(0.01)** | 0.25(0.02) | **0.07(0.03)** |

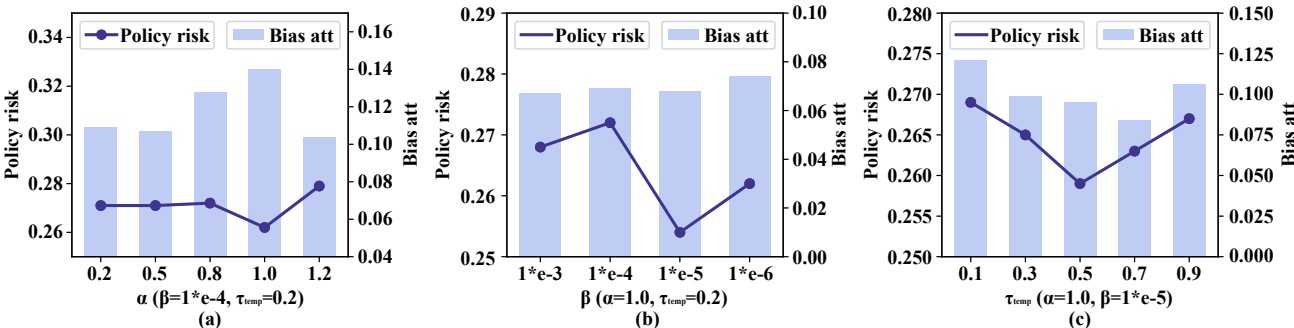

*Figure 6.* ITE estimation performance of our FCCL under different parameters on Jobs dataset.

### D.3. Additional Experiments on Synthetic dataset.

We generate covariates from the multivariate normal distribution $\mathcal{N}\left(\mathbf{0}, \gamma \cdot \sigma^2 \cdot \left[\rho \mathbf{1}_p \mathbf{1}_p^\top + (1-\rho)\mathbf{I}_p\right]\right)$, where the covariance matrix combines an all-ones matrix $\mathbf{1}_p \mathbf{1}_p^\top$ and an identity matrix $\mathbf{I}_p$. The scaling parameter $\gamma \in \{0.4, 0.7, 1.0, 1.2\}$ controls the degree of covariate dispersion, as shown in Figure 7. We sample 800 units with parameters $p = 10$, $\rho = 0.2$, $\sigma^2 = 3$, $\beta_0 = [0.2, ..., 0.2]$, and $\beta_1 = [1.2, ..., 1.2]$. For each $\gamma$, we generate 30 independent datasets, dividing them into training, validation, and test sets with ratios of 63%, 27%, and 10%, respectively. The data generation process is outlined as follows:

$$\mathbf{X}_i \sim \mathcal{N}\left(\mathbf{0}, \gamma \cdot \sigma^2 \cdot \left[\rho \mathbf{1}_p \mathbf{1}_p^\top + (1-\rho)\mathbf{I}_p\right]\right),$$
$$T_i \mid \mathbf{X}_i \sim \text{Bernoulli}\left(1/(1 + \exp\left(-\mathbf{1}_p^\top \mathbf{X}_i\right))\right),$$
$$Y_i^0 = \boldsymbol{\beta_0} \mathbf{X}_i + \xi_i, \quad Y_i^1 = \boldsymbol{\beta_1} \mathbf{X}_i + \xi_i, \quad \xi_i \sim \mathcal{N}(0, 1).$$

Figure 7 presents T-SNE visualizations of the covariates under different values of $\gamma$. At $\gamma = 0.4$, the treated and control groups exhibit relatively concentrated distributions. As $\gamma$ increases to 1.0 and 1.2, the covariate distributions become significantly more dispersed, leading to a greater number of boundary and extreme samples. This trend highlights how larger $\gamma$ values represent greater heterogeneity, providing various scenarios for the individual treatment effect estimation.

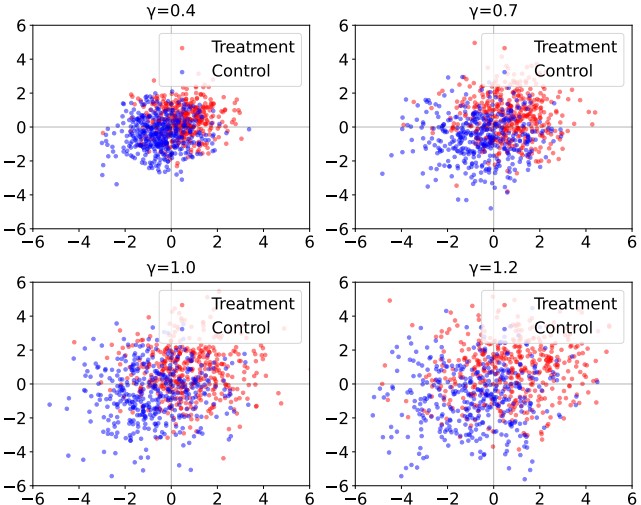

*Figure 7.* T-SNE visualizations of the covariates as $\gamma$ varies.

We report additional experimental results as a supplement to the main text. Table 7 presents the results, where FCCL

*Table 7.* Additional experimental results on Synthetic datasets.

| Method | $\gamma = 0.4$ | | $\gamma = 0.7$ | | $\gamma = 1.0$ | | $\gamma = 1.2$ | |
|---|---|---|---|---|---|---|---|---|
| | $\sqrt{\epsilon_{PEHE}^{within}}$ | $\sqrt{\epsilon_{PEHE}^{out-of}}$ | $\sqrt{\epsilon_{PEHE}^{within}}$ | $\sqrt{\epsilon_{PEHE}^{out-of}}$ | $\sqrt{\epsilon_{PEHE}^{within}}$ | $\sqrt{\epsilon_{PEHE}^{out-of}}$ | $\sqrt{\epsilon_{PEHE}^{within}}$ | $\sqrt{\epsilon_{PEHE}^{out-of}}$ |
| OLS-1 | 8.39(0.38) | 8.41(0.84) | 10.86(0.43) | 10.85(1.34) | 12.89(0.59) | 13.00(1.68) | 14.21(0.59) | 14.32(1.55) |
| OLS-2 | 5.92(0.27) | 5.94(0.60) | 7.64(0.30) | 7.64(0.96) | 9.05(0.40) | 9.13(1.18) | 9.97(0.41) | 10.07(1.11) |
| BART | 4.04(0.22) | 3.40(0.50) | 4.70(0.20) | 4.17(0.60) | 5.36(0.31) | 4.86(0.61) | 5.81(0.26) | 6.14(0.58) |
| KNN | 4.55(0.33) | 5.50(0.62) | 6.48(0.37) | 7.26(0.75) | 7.91(0.38) | 9.08(1.27) | 8.76(0.46) | 10.27(1.07) |
| DML | 4.10(0.26) | 3.96(0.54) | 4.96(0.26) | 5.30(0.62) | 5.72(0.52) | 6.08(0.80) | 7.76(0.62) | 8.46(0.77) |
| CFR-Wass | **2.48(0.05)** | **2.48(0.06)** | 3.73(0.05) | 3.60(0.09) | 4.68(0.07) | 4.72(0.14) | 5.34(0.07) | 5.37(0.14) |
| CFR-MMD | 2.54(0.05) | 2.54(0.06) | 3.75(0.05) | 3.62(0.09) | 4.70(0.07) | 4.74(0.14) | 5.37(0.08) | 5.41(0.14) |
| SITE | 2.68(0.11) | 2.69(0.13) | 4.17(0.16) | 4.25(0.23) | 5.98(0.34) | 6.01(0.38) | 6.19(0.15) | 6.21(0.17) |
| CITE | 2.69(0.04) | 2.71(0.07) | 3.81(0.06) | 3.70(0.10) | 4.68(0.06) | 4.74(0.14) | 5.39(0.07) | 5.41(0.14) |
| GANITE | 4.66(0.03) | 4.69(0.06) | 6.20(0.03) | 6.16(0.07) | 7.32(0.03) | 7.33(0.07) | 8.08(0.03) | 8.11(0.08) |
| ABCEI | 2.73(0.03) | 2.75(0.06) | 3.74(0.04) | 3.57(0.09) | 4.61(0.05) | 4.73(0.13) | 5.19(0.06) | 5.19(0.12) |
| CBRE | 2.91(0.03) | 2.93(0.05) | 4.01(0.04) | 3.85(0.08) | 4.95(0.05) | 5.02(0.12) | 5.77(0.06) | 5.73(0.13) |
| DIGNet | 3.17(0.09) | 3.18(0.11) | 4.09(0.10) | 3.97(0.13) | 5.05(0.10) | 5.10(0.17) | 5.78(0.09) | 5.81(0.15) |
| **FCCL** | 2.56(0.04) | 2.58(0.06) | **3.65(0.05)** | **3.50(0.09)** | **4.40(0.06)** | **4.49(0.13)** | **5.10(0.06)** | **5.12(0.12)** |

achieves smaller $\sqrt{\epsilon_{PEHE}}$ than baselines, especially when covariate distributions become increasingly dispersed (e.g., $\gamma = 1.2$). This demonstrates our method's superior capability in capturing individual-level heterogeneity, especially under high covariate dispersion with sparse boundary samples.

