# OpenReview forum: "Counterfactual Contrastive Learning with Normalizing Flows for Robust Treatment Effect Estimation"
_ICML.cc/2025/Conference — ICML 2025 poster_

### Official Review · Reviewer_Zyty · 2025-03-10

**Overall Recommendation:** 4

**Summary:**

The paper points out that the prediction of the individual treatment effect (ITE) is crucial for personalized therapy planning and proposes a contrastive learning approach (along the lines of SIMCLR) to estimate it. The accuracy of an estimator is measured in terms of the expected squared error of the ITE estimates with respect to the ground-truth. However, ground-truth ITEs are not accessible in practice (since the counter-factual outcome is unknown), and existing sample alignment methods are not good enough for applications with high-dimensional covariates and considerable individual heterogeneity. To circumvent this difficulty, the authors derive a tractable contrastive upper bound for the squared loss that justifies their choice of learning method. Experimental comparisons with several baselines demonstrate the superior behavior of the new method.

Crucially, the authors explain that contrastive training examples cannot be generated by standard data augmentations or adversarial training in data space, because these samples are not sufficiently realistic -- they are too far from the manifold of typical covariate samples. Instead, they propose to first train a normalizing flow to represent the data distribution around the manifold, and then perform a gradient search for contrastive samples in the latent space, where the gradient is calculated from the Jacobian of the decoder. The search for contrastive examples thus stays near the manifold, and the generated counter-factual training data are much more realistic.

EDIT after rebuttal: The authors appropriately addressed my questions and concerns, so I've increased my score.

**Claims And Evidence:**

The claims are largely clear and well supported.

However, I am surprised that debiased ("double") machine learning (e.g. [1]) is neither mentioned as related work nor included in the comparison. In my understanding, debiased ML is a relatively simple method that can estimate ITEs in the absence of randomized controlled experiments, where selection bias of treatment decisions would otherwise contaminate the predictions. I'm curious how the authors asses this possibility.

[1] https://matheusfacure.github.io/python-causality-handbook/22-Debiased-Orthogonal-Machine-Learning.html#non-parametric-double-debiased-ml

**Essential References Not Discussed:**

see above.

**Experimental Designs Or Analyses:**

Evaluation follows standard practices in the field and standard benchmarks. Results appear trustworthy.

It would have been helpful to include an experiment demonstrating that standard data augmentation and adversarial training fail and how the flow-based method fixes this. For example, one could use the Maximum Mean Discrepancy (MMD) to quantify the difference between the distributions of real and synthetic (counter-factual) covariantes. Then, the MMD should be considerably smaller for the proposed method.

The latter experiment could be augmented by an ablation study, showing how ITE results deteriorate when flow-based counter-factual generation is not used.

**Methods And Evaluation Criteria:**

Evaluation follows standard practices in the field and standard benchmarks. Results appear trustworthy.

**Other Comments Or Suggestions:**

none

**Other Strengths And Weaknesses:**

The main weakness of the paper are some minor presentation glitches.

* Definition 3.6 (and repeatedly further down the paper, e.g. equations (6) - (8)): The text refers to "the classification decision k(x)", but k(x) is undefined.

* Lemma 4.1, Theorem 4.2: disc(., .) (the distance in representation space) is never concretely defined, nor are the pros and cons of possible choices discussed. How is it related to the similarity sim(.,.) in equation (9)?

* Second line of section 5.1: possible typo [rho 1 p 1 p'] => [rho 1 p 1^T p' ] ?

**Questions For Authors:**

Please answer the following questions (I'm willing to increase my score):

* relation to debiased ML

* how to fix presentation shortcomings

* experiments demonstrating the superiority of flow-based counter-factual generation

**Relation To Broader Scientific Literature:**

The literature is reviewed well and compared under standard protocols with the proposed method, except for the possible omission of debiased ML mentioned above.

**Theoretical Claims:**

The claims in Lemma 4.1 and Theorem 4.2 are plausible, but I did not check the proofs.

---

> ### Author Rebuttal · Authors · 2025-03-31
>
> We greatly appreciate your high recognition of our work, and are eager to share our thoughts with you.
>
> $\textbf{Q1:}$ Relation to debiased ML.
>
> $\textbf{R1:}$ Thanks for your insightful idea. We would like to clarify that our work primarily focuses on the challenges of deep learning-based causal effect estimation. As debiased machine learning (DML) is a traditional method, we did not explicitly discuss it in the literature review. We will include this discussion in the revision.
>
> Then, we discuss the differences and connections between DML and our FCCL.
>
> First, we explain the different mechanisms that DML and our proposed FCCL use to address confounding bias. DML handles confounding bias through orthogonal residual regression, i.e., $Y_{i}-M_{y}(X_{i}) =\tau(X_{i})(T_{i}-M_{t}(X_{i}))+\epsilon_{i} $, where $M_{y}(X_{i})$ eliminates the influence of the confounder $X$ on $Y$ and $M_{t}(X_{i})$ eliminates the influence of the confounder $X$ on $T$[1]. The way creates statistical independence between treatment and confounder to simulate randomization conditions. Instead, our method captures the characteristics of potential outcomes under different treatments, mitigating distribution shifts through sample-level alignment to emulate RCTs, which better captures individual-level heterogeneity.
>
> Second, the DML way to addressing bias is highly instructive, particularly its unbiasedness. Therefore, we see great potential in incorporating DML into our future work, as R-learners draw inspiration from DML's approach to handling confounding bias, $\tau(\cdot)=argmin_{\tau}\frac{1}{n} \sum_{i=1}^{n} ((Y_{i} -M_{y}(X_{i}))-\tau(X_{i})(T_{i} -M_{t}(X_{i})) )^{2}  $. Thanks again.
>
> Besides, we add comparison experiments with DML (Table1) and the results show that FCCL achieves lower estimation errors.
>
> Table1 ITE estimation errors (std) comparison with DML on IHDP.
> | Method | $\epsilon_{PEHE}^{within}$ | $\epsilon_{ATE}^{within}$ | $\epsilon_{PEHE}^{out-of}$ | $\epsilon_{ATE}^{out-of}$ |
> |--------|----------------------------|----------------------------|-----------------------------|----------------------------|
> | DML    | 2.87(0.09)                 | 0.30(0.05)                 | 2.95(0.14)                  | 0.35(0.03)                 |
> | FCCL   | 0.53(0.04)                 | 0.09(0.01)                 | 0.64(0.07)                  | 0.12(0.02)                 |
>
> $\textbf{Q2:}$ How to fix presentation shortcomings?
>
> $\textbf{R2:}$ Thank you for pointing out the typos, we will correct them in next version.
>
> ( 1 ) We will add a formal definition of $k(x)$ in $\textbf{Definition 3.6}$.
>
> ( 2 ) We thank the reviewer for the suggestion. Indeed, $dis(\cdot ,\cdot )$ denotes the inner product of vectors in the representation space,i.e.,
>     $disc(\Phi^{t}(x),\Phi^{t}(\tilde{x}))=u \sum_{i} b_{i} \sqrt{2-2{{\Phi }^{t=1}({x}\_{i})}^{T}{\Phi }^{t=1}({\tilde{x} }\_{i})}+(1-u)\sum_{i} {a}_{i}\sqrt{2-2{{\Phi }^{t=0}({x}\_{i})}^{T}{\Phi }^{t=0}({\tilde{x} }\_{i})}$. Detailed proofs are provided in Appendix A. $\textbf{Theorem 4.2} $ shows that the estimation error $\epsilon _{PEHE}(h,\Phi )$ is upper bounded by the distance constraints in the representation space. Therefore, we implement the optimization via contrastive loss, specifically by leveraging the cosine similarity $sim(\cdot ,\cdot )$.
>
> ( 3 ) We apologize and will thoroughly go over this for the revision, specifically changing $[\rho \mathbf{1}{p} \mathbf{1}{p}^{\prime}+(1-\rho) \mathbf{I}{p}]$ to $[\rho \mathbf{1}\_{p} \mathbf{1}\_{p}^{\prime}+(1-\rho) \mathbf{I}\_{p}]$, where $\mathbf{1}\_{p}$ denotes the $p$-dimensional all-ones vector and $\mathbf{I}\_{p}$ denotes the identity matrix of size $p$.
>
> $\textbf{Q3:}$ Experiments demonstrating the superiority of flow-based counter-factual generation. For example, one could use the MMD to quantify the difference between the distributions of real and synthetic (counter-factual) covariantes.
>
> $\textbf{R3:}$ Following your valuable suggestion, we use the MMD to quantify the difference between the distributions of factual and counterfactual covariates (Table 2) and the result shows that the MMD is considerably smaller for our FCCL. Besides, we compare the effect of alternative counterfactual generation strategies on ITE estimation error in Table 3 in the main text. These results show the superiority of flow-based counterfactual generation.
>
> Table2 MMD mean (std) comparison on IHDP.
> | Method | grad asc in $\mathcal{X}$ | GAN   | FCCL  |
> |--------|---------------------------|-------|-------|
> | MMD    | 0.13(0.16)                | 0.52(0.55) | 0.09(0.14) |
>
> [1] https://matheusfacure.github.io/python-causality-handbook/22-Debiased-Orthogonal-Machine-Learning.html#non-parametric-double-debiased-ml

---

> > ### Comment · Reviewer_Zyty · 2025-04-03
> >
> > > R1: As debiased machine learning (DML) is a traditional method, we did not explicitly discuss it in the literature review. We will include this discussion in the revision.
> >
> > The regression components of DML can just as well be implemented by neural networks, so it is not restricted to traditional methods :-)
> >
> > Otherwise, your answers appropriately address my concern. Please make sure to revise the paper accordingly.

---

> > > ### Author Response · Authors · 2025-04-04
> > >
> > > We agree with your comment. The regression components of DML can indeed be implemented using any prediction model, including neural networks. We implemented DML with neural networks for the regression components, and the experimental results are presented in the table below. As shown, our FCCL still demonstrates superior performance compared to DML.
> > >
> > > Table1 ITE estimation errors (std) comparison with DML on IHDP.
> > > | Method    | $\epsilon _{PEHE}^{within}$ | $\epsilon _{ATE}^{within}$ | $\epsilon _{PEHE}^{out-of}$ | $\epsilon _{ATE}^{out-of}$ |
> > > |-----------|-----------------------------|----------------------------|-----------------------------|----------------------------|
> > > | DML (RF-based) | 2.87(0.09)                  | 0.30(0.05)                 | 2.95(0.14)                  | 0.35(0.03)                 |
> > > | DML (NN-based) | 2.45(0.12)                  | 0.20(0.05)                 | 2.60(0.14)                  | 0.33 (0.05)                 |
> > > | FCCL      | 0.53(0.04)                  | 0.09(0.01)                 | 0.64(0.07)                  | 0.12(0.02)                 |
> > >
> > > We would like to clarify that our categorization of DML as a "traditional method" was based on how it handles confounding bias through orthogonal residual regression, where DML operates directly on the confounder X. Instead, deep learning methods (also known as representation learning) learn a representation space $\Phi(\cdot)$ to obtain $\phi(X)$, and handle the confounding bias in the representation space.
> > >
> > > We sincerely hope our responses can resolve your concern. In light of these clarifications, we respectfully invite you to consider raising the score.

---

### Official Review · Reviewer_N1DA · 2025-03-13

**Overall Recommendation:** 3

**Summary:**

This paper presents FCCL, an ITE estimation method. FCCL integrates diffeomorphic counterfactual generation and contrastive learning to align treatment and control groups at a fine-grained, sample level, mitigating distribution shifts and approximating RCT randomization. By ensuring realistic counterfactuals and enforcing semantic consistency, FCCL lowers ITE estimation error. Experiments indicate superior performance in heterogeneous and data-scarce settings.

**Claims And Evidence:**

Yes

**Essential References Not Discussed:**

NA

**Experimental Designs Or Analyses:**

- The experiments are valid; although it would have been nice to include more complicated datasets such as ACIC. This is important to evaluate robustness and generalization of the proposed method in large scale settings.
- The evaluation metric “dis” in Fig. 3 is not well-motivated (see section "Questions For Authors” for a specific question).

**Methods And Evaluation Criteria:**

Yes

**Other Comments Or Suggestions:**

NA

**Other Strengths And Weaknesses:**

Strengths:
- The paper provides strong theoretical grounding.
- Innovative combination of normalizing flows (to maintain semantic meaning in counterfactual samples) and contrastive learning (to ensure robust alignment), addressing core challenges in causal inference.

Weaknesses:
- The paper could improve in terms of clarity and flow. Specifically, theory is presented before providing a clear intuition or practical context, which makes it hard to follow the core logic. As a result, readers might struggle with intuitive understanding without first seeing practical motivation.
- The evaluation could benefit from inclusion of more challenging datasets (e.g., ACIC).
- The metric "dis" used in evaluations lacks clear motivation.

**Questions For Authors:**

1. It is stated that the proposed method is specifically useful for “handling sparse boundary samples.” However, neither a proper definition is provided, nor a discussion of why handling them is challenging.
2. Is optimization end-to-end? or are diffeomorphic counterfactuals found first and then the representation and prediction modules are trained?
3. The evaluation metric “dis” (average distance between boundary samples and corresponding class centers) in Fig. 3 is stated to be reflecting sample heterogeneity. However, it’s not mentioned why a smaller dis is desirable.

**Relation To Broader Scientific Literature:**

NA

**Theoretical Claims:**

I looked at Lemma 4.1 and Theorem 4.2. They seem to be correct.

---

> ### Author Rebuttal · Authors · 2025-03-31
>
> We thank the reviewer for their time and for recognizing the importance of heterogeneous treatment effect estimation. Please see below answers to the questions.
>
> $\textbf{Q1:}$ Provide a proper definition of boundary samples and discuss why handling them is challenging.
>
> $\textbf{R1:}$ We thank the reviewer for consideration. We provide a formal definition of boundary samples in Appendix D.1, and will include this definition in the main text in the revision.
>
> Furthermore, we analyze the challenges of handling boundary samples from two perspectives:
>
> (1) Many scenarios demand flexible investigation of effect heterogeneity across individuals. However, boundary samples, due to their distinctive characteristics, reside in low-probability density regions distant from class centroids, which poses challenges for accurate ITE estimation.
>
> (2) Existing methods, such as MMD, minimize the distribution discrepancy between treated and control groups by aligning their mean representations $\frac{1}{m\_1}\textstyle\sum_{i=1}^{m\_1}{\phi}(x\_{i}^{t})$ and $\frac{1}{m\_2}\textstyle\sum_{j=1}^{m\_2}{\phi}(x_{j}^{c})$. However, such methods overlook samples near the boundaries of the treatment and control distributions.
>
> Our method achieves robust performance via sample-level alignment, especially in scenarios where individual differences are significant. Besides, we also provide boundary sample analysis by listing the estimation errors of the first five samples from Table 4 in Appendix D, as shown in Table 1. More detailed results are presented in Appendix D.
>
> Table1 The ITE estimation error $|\epsilon_{ITE}|$ of boundary samples on IHDP.
> | Sample | CFR-MMD | FCCL   |
> |--------|---------|--------|
> | 1      | 4.2264  | 0.2144 |
> | 2      | 3.0977  | 0.0594 |
> | 3      | 0.0337  | 0.0805 |
> | 4      | 6.1246  | 0.6418 |
> | 5      | 2.1014  | 0.2263 |
>
> $\textbf{Q2:}$ Is optimization end-to-end? or are diffeomorphic counterfactuals found first and then the representation and prediction modules are trained?
>
> $\textbf{R2:}$ Our approach falls into the latter: we first generate the diffeomorphic counterfactuals, and then train the representation and prediction modules.
>
> $\textbf{Q3:}$ Why the smaller evaluation metric “dis” is desirable?
>
> $\textbf{R3:}$ A smaller "dis" value indicates that dispersion of samples in the latent representation space is reduced, which further reflects a smaller distance between boundary samples and their corresponding class centers. This shows that there are fewer boundary samples, enabling better model fitting and exhibiting lower ITE estimation bias, as demonstrated by our results. We will make this point clearer in the next version.
>
> $\textbf{Q4 (Weaknesses2):}$ The evaluation could benefit from inclusion of more challenging datasets (e.g., ACIC).
>
> $\textbf{R4:}$ Thank you for your valuable suggestions. For validation, we additionally evaluate our method against several representative baselines on ACIC (Table 2). We observe that our method still outperforms the baselines.
>
> Table2 Within-sample and out-of-sample mean (std) for the metrics on ACIC.
> | Method  | $\epsilon_{PEHE}^{within}$ | $\epsilon_{ATE}^{within}$ | $\epsilon_{PEHE}^{out-of}$ | $\epsilon_{ATE}^{out-of}$ |
> |---------|----------------------------|----------------------------|-----------------------------|----------------------------|
> | CFR-MMD | 1.70(0.38)                 | 0.29(0.12)                 | 2.36(0.59)                  | 0.30(0.12)                 |
> | SITE    | 1.71(0.39)                 | 0.38(0.14)                 | 2.33(0.58)                  | 0.39(0.15)                 |
> | ABCEI   | 1.93(0.46)                 | 0.17(0.07)                 | 2.49(0.64)                  | 0.18(0.07)                 |
> | CBRE    | 1.69(0.38)                 | 0.12(0.05)                 | 2.31(0.56)                  | 0.14(0.06)                 |
> | DIGNet  | 1.66(0.35)                 | 0.23(0.07)                 | 2.30(0.54)                  | 0.24(0.07)                 |
> | FCCL    | 1.51(0.40)                 | 0.16(0.05)                 | 2.16(0.56)                  | 0.17(0.05)                 |

---

> > ### Comment · Reviewer_N1DA · 2025-04-03
> >
> > I thank the authors for addressing my comments.
> >
> > Please make sure to clarify R2 in the paper.
> > BTW, would it be possible to design the algorithm to be end-to-end?

---

> > > ### Author Response · Authors · 2025-04-04
> > >
> > > Thank you for your suggestion. We aim to generate semantically meaningful counterfactuals, but an end-to-end design would likely introduce non-negligible interference gradients into training and thus impair the quality of counterfactual generation. Nevertheless, we will explore better approaches to design the algorithm to be end-to-end.

---

### Official Review · Reviewer_HBA5 · 2025-03-14

**Overall Recommendation:** 3

**Summary:**

The paper introduces Flow-based Counterfactual Contrastive Learning (FCCL), a novel approach for Individual Treatment Effect (ITE) estimation that integrates normalizing flows for realistic counterfactual generation and contrastive learning for fine-grained sample alignment. It derives a theoretical generalization-error bound linking ITE estimation error to factual prediction error and representation distances. Empirical evaluations on synthetic, semi-synthetic (IHDP), and real-world (Jobs) datasets demonstrate FCCL’s superior performance

**Claims And Evidence:**

Claim 1: FCCL generates realistic counterfactuals that adhere to the data manifold.

Supported. Through the use of normalizing flows, which enforce structure on counterfactual transformations. The authors present visualization results showing that FCCL maintains sample-level semantic consistency better than baseline methods.

Claim 2: FCCL reduces ITE estimation error via sample-level alignment.

Partially supported. While contrastive learning improves alignment, the empirical evidence (particularly in Tables 1 and 2) primarily shows improvement in error metrics rather than a direct measure of sample alignment. Additional ablation studies isolating the contrastive loss would be beneficial.

Claim 3: The proposed theoretical generalization-error bound justifies FCCL’s effectiveness.

Supported. The bound is mathematically derived (Theorem 4.2) and aligns with the contrastive loss objective. However, empirical validation of whether this bound holds in practice is not explicitly tested.

Claim 4: FCCL significantly outperforms state-of-the-art baselines.

Mostly supported. The results show FCCL achieving the best ϵ_PEHE and ϵ_ATE scores across datasets. However, the magnitude of improvement varies, and for some cases (e.g., IHDP out-of-sample ATE error), the advantage is marginal.

**Essential References Not Discussed:**

The paper extensively cites prior work but does not discuss alternatives to normalizing flows for counterfactual generation, such as energy-based models (Du et al., 2021) or diffusion-based generative approaches (Song et al., 2021).

**Experimental Designs Or Analyses:**

The experiment design is generally well-structured, with appropriate train-test splits and multiple trials.

Missing Analyses: The impact of hyperparameters (especially temperature τ in contrastive loss) is not explored. Additionally, no robustness checks (e.g., sensitivity to noise or dataset shifts) are provided.

**Methods And Evaluation Criteria:**

The use of benchmark datasets (IHDP, Jobs, synthetic data) is appropriate for treatment effect estimation.
Baselines (OLS, CFR, GANITE, ABCEI, etc.) are well-chosen, covering both traditional and deep learning-based ITE estimation methods.
Evaluation metrics (ϵ_PEHE, ϵ_ATE, ATT) are standard, but additional fairness metrics (e.g., subgroup fairness or bias analysis) could strengthen the evaluation.

The use of latent space visualizations (Figure 3) is insightful, but further quantitative measures of alignment (e.g., KL divergence or propensity score matching quality) would reinforce the claims.

**Other Comments Or Suggestions:**

no

**Other Strengths And Weaknesses:**

no

**Questions For Authors:**

1) How does FCCL compare to diffusion-based counterfactual generators? Diffusion models are gaining traction in structured data. Would FCCL’s theoretical framework extend to them?
2) How stable is FCCL under different hyperparameter choices? Particularly, how does the contrastive loss temperature τ affect performance?
3) Can FCCL generalize to continuous treatments? The paper focuses on binary treatments, but many real-world applications require handling continuous interventions.

**Relation To Broader Scientific Literature:**

FCCL extends prior work on representation learning for ITE estimation (e.g., CFR, SITE, CITE) to contrastive learning for causal inference, aligning with recent trends in self-supervised learning for structured data.
The approach could be adapted to multi-treatment and continuous intervention settings, similar to recent developments in continuous treatment estimation (e.g., Kazemi & Ester, 2024).

**Theoretical Claims:**

Correctness of Proofs: The theoretical derivations appear correct and align with existing literature on treatment effect bounds (e.g., Shalit et al., 2017).

Missing Considerations: The assumptions regarding the invertibility of representations (Φ) and the geodesic distance formulation could be further justified. Additionally, the bound does not account for sample sparsity effects, which are critical in real-world datasets.

---

> ### Author Rebuttal · Authors · 2025-03-31
>
> Thank you for your insightful suggestions. We have addressed the comments related to the counterfactual generation and model robustness evaluation. Please see our responses below.
>
> $\textbf{Q1:}$ How does FCCL compare to diffusion-based counterfactual generators?
>
> $\textbf{R1:}$ We thank the reviewer for the question. We add experiments on diffusion-based counterfactual generation (Table1), which shows performance comparable to FCCL, but with slightly inferior result on $\epsilon _{PEHE}^{out-of}$. This may be due to the noise-driven mechanism of the diffusion model [1], which causes counterfactual to deviate from the sample semantic space, especially in the out-sample cases. In contrast, the flow-based model ensures that counterfactuals reside on the same manifold as the original instances, which generates both meaningful and reliable counterfactuals, making FCCL more reliable for individual-level treatment effect predictions. We will include this discussion in the revision.
>
> Table1 ITE estimation errors (std) with different generation methods on IHDP.
> | Method    | $\epsilon _{PEHE}^{within}$ | $\epsilon _{ATE}^{within}$ | $\epsilon _{PEHE}^{out-of}$ | $\epsilon _{ATE}^{out-of}$ |
> |-----------|-----------------------------|----------------------------|-----------------------------|----------------------------|
> | diffusion-based | 0.54(0.05)   | 0.09(0.01)   | 0.72(0.10)  | 0.12(0.03) |
> | FCCL     | 0.53(0.04)  | 0.09(0.01)   | 0.64(0.07)  | 0.12(0.02) |
>
> $\textbf{Q2:}$ How does the contrastive loss temperature $\tau $ affect performance?
>
> $\textbf{R2:}$ We perform sensitivity analysis focusing on the temperature coefficient $\tau$ (see Table 2). We observe minimal variation in performance with respect to $\tau $, which demonstrates our model's general robustness.
>
> Table2 ITE estimation errors (std) with different values of $\tau $ on IHDP.
> | $\tau$ | $\epsilon_{PEHE}^{within}$ | $\epsilon_{ATE}^{within}$ | $\epsilon_{PEHE}^{out-of}$ | $\epsilon_{ATE}^{out-of}$ |
> |--------|----------------------------|----------------------------|-----------------------------|----------------------------|
> | 0.1  | 0.54(0.04)   | 0.11(0.02)  | 0.64(0.06)  | 0.13(0.02) |
> | 0.3  | 0.56(0.04)   | 0.10(0.02)  | 0.70(0.08)  | 0.14(0.02) |
> | 0.5  | 0.58(0.06)   | 0.10(0.02)  | 0.76(0.16)  | 0.11(0.02) |
> | 0.7  | 0.61(0.06)   | 0.10(0.02)  | 0.79(0.15)  | 0.11(0.02) |
> | 0.9  | 0.56(0.04)   | 0.10(0.01)  | 0.69(0.07)  | 0.12(0.02) |
>
> $\textbf{Q3:}$ Can FCCL generalize to continuous treatments?
>
> $\textbf{R3:}$ This is a valuable point. FCCL currently focuses on binary treatments, we can extend it to continuous treatments in future work. The most direct approach would involve discretizing the continuous treatment variable[2]. Specifically, we can split treatments into $E$ heads, each assigned a dose level from the range$[a, b]$, which is divided into $E$ equal intervals of width $(b-a)/E$. We will explore better approaches to handle continuous treatments directly.
>
> $\textbf{Q4 (Claim2):}$ Additional ablation studies isolating the contrastive loss would be beneficial.
>
> $\textbf{R4:}$ We discussed the impact of contrastive loss, thank you for going into Figure 4 in the main text.
>
> $\textbf{Q5 (Claim3):}$ Empirical validation of whether this ITE bound holds in practice is not explicitly tested.
>
> $\textbf{R5:}$ Our empirical analysis demonstrates that the proposed ITE bound effectively guides ITE model training and becomes tighter than the bound proposed in CFR as iterations increase under identical conditions[3] (Table 3).
>
> Table3 Generalization-error bound comparison on IHDP.
> | Iterations | 400 | 800 | 1200 | 1600 | 2000 |
> |------------|---------|---------|---------|---------|---------|
> | CFR  | 13.24 | 10.84 | 10.06 | 9.64 | 9.91 |
> | FCCL | 8.54 | 3.74 | 2.93 | 2.77 | 2.67 |
>
> $\textbf{Q6 (Methods And Evaluation Criteria):}$ Further quantitative measures of alignment would reinforce the claims (e.g., KL divergence).
>
> $\textbf{R6:}$ Thank you for your suggestion. We add KL divergence comparisons between treatment and control distributions for two typical methods (see Table 4), which demonstrates that our FCCL can better address distribution shifts. We will include this metric in Figure 3 in the next version.
>
> Table4 KL divergence mean (std) comparison on IHDP.
> | Method | CFR  | ABCEI | FCCL |
> |--------|-----------|-----------|-----------|
> | KL | 0.14(0.08) | 0.20(0.30) | 0.09(0.05) |
>
> [1] Kotelnikov Akim, et al. Tabddpm: Modelling tabular data with diffusion models. In International Conference on Machine Learning, pp. 17564–17579. PMLR, 2023.
>
> [2] Schwab et al. Learning counterfactual representations for estimating individual dose-response curves. Proceedings of the AAAI Conference on Artificial Intelligence, pp. 5612-5619, 2020.
>
> [3] Shalit, U., et al. Estimating individual treatment effect: generalization bounds and algorithms. In International Conference on Machine Learning, pp. 3076–3085. PMLR, 2017.

---

### Official Review · Reviewer_PLY3 · 2025-03-21

**Overall Recommendation:** 4

**Summary:**

The paper proposes FCCL  framework for the ITE estimation.  The proposed method can generate realistic counterfactuals by leveraging normalizing flows to ensure adherence to the data manifold, preserving semantic similarity to factual samples. The authors also derive a new generalization bound connecting ITE estimation error to factual prediction errors and representation distances between factual-counterfactual pairs, providing theoretical grounding for their proposed sample-level alignment method.

**Claims And Evidence:**

Yes

**Essential References Not Discussed:**

Some papers also discuss the new upper bound of PEHE. Can include them in the literature review.

**Experimental Designs Or Analyses:**

The experimental design and comparisons are proper.

**Methods And Evaluation Criteria:**

Yes

**Other Comments Or Suggestions:**

None.

**Other Strengths And Weaknesses:**

Strengths:

The motivation, presentation, and experimental comparisons are good.

Weaknesses:

I only have one concern about the theoretical parts. One of the key contributions is that the authors claim they propose a new ITE error bound for their sample alignment method. However, anyone can propose an ITE error bound and minimize such a bound to learn the ITE model. The point is, how can you make sure the proposed ITE bound is tight or not? If the bound is too loose, such a theoretical bound may give bad guidance for training the ITE model.

I thus suggest authors give some theoretical or empirical evidence to show that the bound is tight (at least tighter than the bound proposed in CFRWASS)

**Questions For Authors:**

Please see Strengths And Weaknesses

**Relation To Broader Scientific Literature:**

ITE model development.

**Theoretical Claims:**

Yes.

---

> ### Author Rebuttal · Authors · 2025-03-31
>
> We appreciate your time and your thoughtful and encouraging comments. We hope our responses can resolve your concern.
>
> $\textbf{Q1:}$ I suggest authors give some theoretical or empirical evidence to show that the bound is tight (at least tighter than the bound proposed in CFR).
>
> $\textbf{R1:}$ Following your helpful suggestion, we conduct an experimental analysis to compare the generalization error bounds of our proposed FCCL (Equation (14) in Appendix A) with the CFR bound (Equation (8) in Appendix A.2, [1]) under the same conditions.
>
> As shown in Table 1, we track the generalization error bounds at different iterations. The results demonstrate that as the number of iterations increases, FCCL achieves a significantly tighter bound than that of CFR.
>
> Table1 Generalization-error bound comparison on IHDP.
> | Iterations | 400      | 800      | 1200     | 1600     | 2000     |
> |------------|----------|----------|----------|----------|----------|
> | CFR | 13.24 | 10.84 | 10.06 | 9.64 | 9.91 |
> | FCCL | 8.54 | 3.74 | 2.93 | 2.77 | 2.67 |
>
> In addition, we would like to clarify that our FCCL provides a different theoretical perspective from CFR. Our generalization error bound links ITE estimation error to the generalization error of factual predictions and representation distances, motivating our focus on minimizing these distances via sample-level alignment. Instead, the bound of CFR shows ITE estimation error can be reduced by the difference between the treated and control distributions.
>
> [1] Shalit, U., et al. Estimating individual treatment effect: generalization bounds and algorithms. In International Conference on Machine Learning, pp. 3076–3085. PMLR, 2017.

---

### Decision · Program_Chairs · 2025-05-01

**Decision:**

Accept (poster)

**Comment:**

The reviewers generally appreciated the work in combining normalizing flows and contrastive learning to better estimate individual treatment effects (ITE). They generally felt that both the theory and experiments were useful and were explained well.

The authors should aim to include several points of clarification and explanation brought up during the review process in the final version.